# Associations of PFAS and OH-PCBs with risk of multiple sclerosis onset and disability worsening

Aina Vaivade[1], Ida Erngren [1], Henrik Carlsson[1], Eva Freyhult [2], Payam Emami Khoonsari[1,3], Yassine Noui [4], Asma Al-Grety[1], Torbjörn Åkerfeldt[1], Ola Spjuth [5], Valentina Gallo[6], Anders Olof Larsson[1], Ingrid Kockum [7,8,9], Anna Karin Hedström [7], Lars Alfredsson [7,10,11], Tomas Olsson[7,8,9], Joachim Burman [4] & Kim Kultima [1] ✉

Exposure to per- and polyfluorinated substances (PFAS) and hydroxylated polychlorinated biphenyls (OH-PCBs) is associated with adverse human health effects, including immunosuppression. It is unknown if these substances can affect the course of autoimmune diseases. This study was based on 907 individuals with multiple sclerosis (MS) and 907 matched controls, where the MS cases were followed longitudinally using the Swedish MS register. We demonstrate sex- and disease-specific differences in serum PFAS concentrations between individuals with MS and controls. Moreover, two OH-PCBs (4-OH-CB187 and 3-OH-CB153) are associated with an increased risk of developing multiple sclerosis, regardless of sex and immigration status. With a clinical follow-up time of up to 18 years, an increase in serum concentrations of perfluorooctanoic acid (PFOA), perfluorooctane sulfonic acid (PFOS), and perfluorodecanoic acid (PFDA) decreases the risk of confirmed disability worsening in both sexes, as well as perfluoroheptanesulfonic acid (PFHpS) and perfluorononanoic acid (PFNA), only in males with MS. These results show previously unknown associations between OH-PCBs and the risk of developing MS, as well as the inverse associations between PFAS exposure and the risk of disability worsening in MS.

Multiple sclerosis (MS) is the most common autoimmune inflammatory neurological disease among young adults, affecting more than 2.9 million people worldwide[1–3]. Onset is typically in early adulthood, and with time, patients tend to develop significant disability. Approximately 80–90% of patients with MS are initially diagnosed with relapsing-remitting MS (RRMS), characterized by intermittent periods of symptom worsening, known as relapses, followed by partial or complete resolution of the symptoms. This is contrasted with the 5–15% of

[1]Department of Medical Sciences, Clinical Chemistry, Uppsala University, Uppsala, Sweden. [2]Department of Cell and Molecular Biology, Uppsala University, Uppsala, Sweden. [3]Department of Biochemistry and Biophysics, National Bioinformatics Infrastructure Sweden, Science for Life Laboratory, Stockholm University, Stockholm, Sweden. [4]Department of Medical Science, Neurology, Uppsala University, Uppsala, Sweden. [5]Department of Pharmaceutical Biosciences, Uppsala University, Uppsala, Sweden. [6]Department of Sustainable Health, Campus Fryslân, University of Groningen, Leeuwarden, the Netherlands. [7]Department of Clinical Neuroscience, Karolinska Institutet, Solna, Sweden. [8]The Karolinska Neuroimmunology & Multiple Sclerosis Centre, Centrum for Molecular Medicine, Karolinska University Hospital, Solna, Sweden. [9]Academic Specialist Center, Stockholm 113 65, Sweden. [10]Institute of Environmental Medicine, Karolinska Institutet, Solna, Sweden. [11]Centre for Occupational and Environmental Medicine, Region, Stockholm, Sweden. ✉e-mail: Kim.Kultima@medsci.uu.se

patients who have primary progressive MS (PPMS), with gradually worsening disability from onset, without relapses[1,4]. A significant portion of patients with RRMS will later also develop gradually worsening disability without relapses, referred to as secondary progressive MS (SPMS).

Environmental and genetic factors have been identified as risk factors for the development of MS, most notably female sex, Epstein-Barr virus infection, smoking, adolescent obesity, and certain HLA-associated genetic variants[1,2,5,6]. Moreover, low vitamin D levels have also been associated with an increased risk of MS, however some uncertainty remains[7,8]. Despite continued advances in these areas, these known risk factors do not account for the entirety of MS risk[9]. Moreover, the effect of known risk factors on disease severity and the development of progressive disease is poorly understood[10,11].

Per- and polyfluoroalkyl substances (PFAS) are a large family of manufactured compounds that exhibit marked hydrophobic and oleophobic properties. They have been used in many industrial and commercial applications since their introduction in the 1940s. The compelling properties that drive their use also result in an extreme resistance to environmental degradation. This, coupled with their pervasive use, has led to their widespread detection as environmental contaminants and in human biofluids. Humans' primary PFAS exposure sources are consumer products, diet, and contaminated drinking water[12]. There is a growing concern that chronic exposure to these endocrine-disrupting chemicals may have adverse effects related to inflammation and immunity[12–18].

Several studies have reported on the immunosuppressive effects of PFAS, mainly perfluorooctanoic acid (PFOA) and perfluorooctanesulfonic acid (PFOS), reflected by an increase in common infectious diseases and decreased vaccine antibody levels in children[14,19–21]. Similar associations have also been observed for perfluorohexanesulfonic acid (PFHxS) and perfluorodecanoic acid (PFDA)[22]. However, less consistent findings have been observed in the adult population[23–26]. Moreover, the current knowledge and findings of PFAS associations with autoimmune disease are scarce and, at large, often have contradicting results in humans (reviewed in 2023)[14]. PFOA exposure has been associated with an increased risk of developing ulcerative colitis[27]. In addition, a recent study demonstrated an association between higher PFAS concentrations and a decreased risk of developing rheumatoid arthritis[28]. To this date, no risk analysis has been performed concerning PFAS exposure and the risk of developing MS. A cross-sectional study examining the relationship between autoimmune disease and estimated past exposure levels of perfluorooctanoic acid (PFOA), a common PFAS, demonstrated no significant association between PFOA levels and the incidence of MS[27]. However, the PFOA concentrations were estimated from intake of contaminated drinking water, assuming low background exposure, and not measured. Additionally, a case-control study showed lower concentrations of PFOA in individuals with MS compared with controls, suggesting that PFOA exposure was not an important risk factor for MS, but no risk analysis was performed[29]. To date, there are no studies investigating whether concentrations of PFAS have any role in disease progression in MS.

Polychlorinated biphenyls (PCBs), another type of endocrine-disrupting chemicals, are well-established as carcinogens and are known to have other harmful effects on human health, including developmental toxicity, neurotoxicity, and immunosuppression. Consequently, many countries have restricted or banned their production and use since the 1970s, though they persist in the environment[15,16]. In humans, hydroxylation of PCBs can occur, resulting in OH-PCBs, and the toxicity of PCBs has been suggested to originate partly from their hydroxylated metabolites. Moreover, OH-PCBs have a range of toxic traits not observed in parent PCBs, such as inhibition of mitochondrial respiration, disruption of thyroid hormone activity and estrogenic activity, and generation of reactive oxygen species[16].

The primary aim of this matched case-control study was to investigate the concentrations of PFAS and OH-PCB compounds in individuals recently diagnosed with MS and their matched controls (HC), and if there could be an association with the risk of developing MS. Additionally, we aimed to explore whether the concentrations of PFAS and OH-PCB compounds influenced the course of the disease. We investigated the association between these chemicals and vitamin D3 levels and explored if the presumed protective effects of vitamin D3 are linked to PFAS and OH-PCB exposure.

Here, we show a significant association between OH-PCBs and an increased risk of developing MS independent of sex and immigration status. Moreover, we demonstrate sex- and disease-specific concentration differences in serum PFAS and OH-PCB concentration between individuals with MS and HC. Finally, with up to 18 years of clinical follow-up time, we demonstrate an association between increased serum concentrations of certain PFAS compounds and decreased risk of confirmed disability worsening in both sexes, and some additional PFAS compounds in males only. This study reveals valuable insights into PFAS and OH-PCBs association with MS and confirmed disability worsening.

## Results

The clinical and lifestyle factors of the study population are presented in Table 1. In total, 907 MS and 907 matched HC subjects (matched on age, sex, and county) from six major healthcare regions in Sweden were included in this study, with 678 (74.8%) females and 229 (25.2%) males in each group. Out of the 907 MS subjects, 801 had a relapsing-remitting disease course, and 106 had a progressive disease course when the blood sample was taken.

The MS cases were followed up to 18 years in the Swedish MS registry, where clinical information and treatment are registered[30]. The median Expanded disability status scale (EDSS) score for individuals with RRMS at sample collection was $1.5 \pm 1.2$ for females and $2.0 \pm 1.6$ for men. For individuals with PMS, the median EDSS was $3.5 \pm 1.5$ for females and $3.5 \pm 1.6$ for men. Further, out of the 801 individuals with RRMS, 89 transitioned to SPMS during the follow-up time. The time from sample time point to SPMS transition for females was $4.2 \pm 3.0$ years and for males $4.2 \pm 2.7$ years (Table 1).

In total, 31 different compounds were analyzed in the serum samples using targeted analysis, including 24 PFAS compounds and seven OH-PCBs. The targeted compounds were selected based on environmental relevance, commercial availability, and their detectability in serum using liquid chromatography-mass spectrometry. The 24 targeted PFAS included common compounds from the most commonly studied PFAS classes, including perfluoroalkyl carboxylic acids ($n = 11$), perfluoroalkylsulfonates ($n = 7$), fluorotelomer sulfonates (FTS) ($n = 3$), perfluorooctanesulfonamidoacetic acids ($n = 2$) as well as the single compound perfluoro-1-octanesulfonamide (FOSA). The seven targeted OH-PCBs included several of the most predominant OH-PCB congeners in human blood, e.g., 2,2′,3,4′,5,5′,6-heptachloro-4-biphenylol (4-OH-CB187)[31]. Among the 31 analyzed compounds, 20 were detected in >20% of HC and MS subjects, and 11 of them were detected in >90% of the samples. Values below the detection limit were imputed with half the lowest measured concentration for each compound. The mean concentrations of each compound in HC, RRMS, and PMS subjects are presented in Supplementary Table S1. It is important to note that exposure to these compounds is not independent of each other, and humans are generally exposed to multiple PFAS or OH-PCBs rather than single compounds.

### Collection year impacts PFAS and OH-PCB concentrations
Many countries have banned PCBs since the 1970s[15,16], and industries have been phasing out several PFAS in the last years (early 2000s)[22]. Therefore, we investigated if serum PFAS and OH-PCB concentrations varied depending on the year the blood sample was taken. As

**Table 1 | General and clinical characteristics for the epidemiological investigation of multiple sclerosis (EIMS) cohort, n = 1814**

| Characteristic | HC | RRMS | PMS |
|---|---|---|---|
| N | 907 | 801 | 106 |
| Sex (F/M) | 678 (74.8%)/229 (25.2%) | 613 (76.5%)/188 (23.5%) | 65 (61.3%)/41 (38.7%) |
| Age: Mean ± SD, F/M | 41.0 ± 11.1/41.3 ± 11.2 | 39.1 ± 10.4/40.0 ± 10.1 | 51.9 ± 8.8 / 51.2 ± 10.1 |
| BMI at sample collection: Mean ± SD, F/M | 24.8 ± 4.6 / 25.9 ± 3.3 | 25.1 ± 5.2 / 25.2 ± 4.1 | 25.1 ± 4.5 / 25.0 ± 2.8 |
| BMI at the age of 20: Mean ± SD, F/M | 21.7 ± 3.4 / 23.1 ± 2.9 | 22.4 ± 4.0 / 23.3 ± 3.3 | 20.8 ± 4.0 / 22.9 ± 3.5 |
| Disease duration | | | |
| Months: Mean ± SD, F/M | - | 8.9 ± 17.4 / 8.8 ± 15.3 | 13.6 ± 39.7 / 10.5 ± 14.2 |
| EDSS* | | | |
| No.F/M | - | 375/119 | 39/18 |
| Mean ± SD, F/M | - | 1.5 ± 1.2 / 2.1 ± 1.6 | 3.8 ± 1.5 / 3.7 ± 1.6 |
| Median ±SD, F/M | - | 1.5 ± 1.2 / 2.0 ± 1.6 | 3.5 ± 1.5 / 3.5 ± 1.6 |
| Transition from RRMS to SPMS | | | |
| % F/M | - | 66.3 %/33.7 % | - |
| Time to transition (years): Mean ± SD, F/M | - | 4.2 ± 3.0/4.2 ± 2.7 | - |
| Treatment | | | |
| % undergoing treatment | - | 10.1% | 15.1% |
| Smoking | | | |
| Regular: % Yes/No/NA | 16.5/83.2/0.2% | 23.7/69.5/6.7% | 17.9/76.4/5.7% |
| Regular: Mean ± SD (years), F/M | 24.8 ± 14.0/22.5 ± 15.1 | 22.0 ± 9.8/21.0 ± 11.4 | 33.5 ± 6.7/32.8 ± 5.0 |
| Irregular: % Yes/No/NA | 2.2/ 97.8/0% | 3.7/96.3/0% | 4.7/ 95.3/0% |
| Irregular: Mean ± SD (years), F/M | 10.3 ± 9.1/11 ± 9.8 | 12.4 ± 11.0/11.4 ± 8.2 | 12.7 ± 12.4/10.5 ± 14.8 |
| Passive smoking (indoors): % Yes/No/NA | 3.1/96.6/0.3% | 4.1/89.2/6.7% | 6.6/87.7/5.7% |
| Passive smoking (indoors): Mean ± SD (years), F/M | 18.8 ± 21.5/19.4 ± 11.7 | 30.8 ± 14.8/19.8 ± 15.4 | NA** /27.3 ± 16.4 |
| Swedish snuff | | | |
| % Yes/No/NA | 4.2/85.9/9.9% | 8.0/80.1/11.9% | 11.3/81.1/ 7.5% |
| Years: Mean ± SD, F/M | 6.8 ± 4.4 / 19.8 ± 9.0 | 9.8 ± 8.2/14.2 ± 9.9 | 3.0 ± 2.8/21.6 ± 15.9 |
| Alcohol consumption*** | | | |
| % Yes/No/NA | 73.4/26.5/0.1% | 64.7/28.7/ 6.6% | 65.1/ 29.2/ 5.7% |
| Volume (cl): Mean ± SD, F/M | 2.4 ± 13.9/6.4 ± 14.2 | 1.4 ± 5.1/3.9 ± 9.0 | 0.8 ± 3.4/4.9 ± 9.1 |
| Infectious mononucleosis | | | |
| % Yes/No/NA | 21.8/77.8/0.3% | 23.5/69.4/7.1% | 22.6/71.7/5.7% |
| Childbirths | | | |
| % Yes/Mean ± SD (number of childbirths) | 71.7/2.1 ± 1.2 | 61.3%/2.1 ± 0.8 | 72.3%/2.0 ± 0.9 |
| Sun exposure | | | |
| % High/Medium/Low /NA | 31.8/39.4/28.8/0% | 20.6/32.7/35.0/11.7% | |

The cohort consists of matched controls (HC), individuals with the diagnosis of relapsing-remitting multiple sclerosis (RRMS), and progressive multiple sclerosis (PMS) at the time of inclusion.

*F* female, *M* male, *BMI* body mass index (kg/m²), *EDSS* expanded disability status scale, *SD* standard deviation.

*Description of EDSS measurements for individuals with an EDSS within three months from blood sample collection.

**Only two females with PMS described that they had been exposed to passive smoking, however, the duration was not specified.

***Alcohol consumption the week prior to sample collection has been recalculated into volume (cl) consumed 40% alcohol.

expected, PFAS and OH-PCB concentrations significantly decreased with the sample collection year, where higher concentrations were detected in blood samples taken at the beginning of this study (Supplementary Fig. S1). Thus, illustrating how the year of sample collection significantly impacts serum concentration. In addition, higher concentrations were observed in males compared to females for the three long-chain PFAS (perfluoroheptanesulfonic acid (PFHpS), PFOA, and PFOS) (Fig. 1).

**Sex and age impact PFAS and OH-PCB concentrations**

Despite the decreasing concentrations observed during the sample collection year, we can still observe that these compounds accumulate in the body. Age significantly affected the estimate for 15 PFAS, and all three OH-PCB compound concentrations increased with age (Supplementary Table S2 and Fig. S2), as exemplified by Fig. 2. Serum 1H,1H,2H,2H-perfluoro-1-decanesulfonic acid (8:2) (8:2 FTS) and perfluorobutanesulfonic acid (PFBS) concentrations were not affected by

age. Moreover, PFHxS ($P = 1.37e-9$), PFHpS ($P = 5.30e-10$), PFOA ($P = 2.54e-4$), PFOS ($P = 9.75e-5$), and 4-OH-CB187 ($P = 0.045$) demonstrated significantly higher concentrations in males, whilst perfluorotridecanoic acid (PFTrDA) ($P = 0.026$) and 8:2 FTS ($P = 0.017$) were higher in females (Fig. 2, Supplementary Fig. S3, and Table S2). Childbirth also impacted PFAS concentrations, where females who had given birth had significantly lower PFAS concentrations (perfluoropentanesulfonic acid (PFPeS); $P = 1.23e-03$, perfluoroheptanoic acid (PFHpA); $P = 8.45e-4$, PFHxS; $P = 3.13e-7$, PFHpS; $P = 1.10e-7$, PFOA; $P = 2.98e-14$, PFOS; $P = 3.39e-3$, perfluorononanoic acid (PFNA); $P = 1.11e-5$, and perfluorododecanoic acid (PFDoA); $P = 0.014$) compared to females who had not given birth. Two OH-PCBs (2,2',4,4',5,5'-,Hexachloro-3-biphenylol (OH-CB153) and 2,3,3',4',5-Pentachloro-4-biphenylol (OH-CB107)) also demonstrated significantly ($P = 0.028$ and $P = 7.20e-3$) lower concentrations in females who had given birth compared to females who had not given birth. As expected, this difference was not observed between males with and without biological

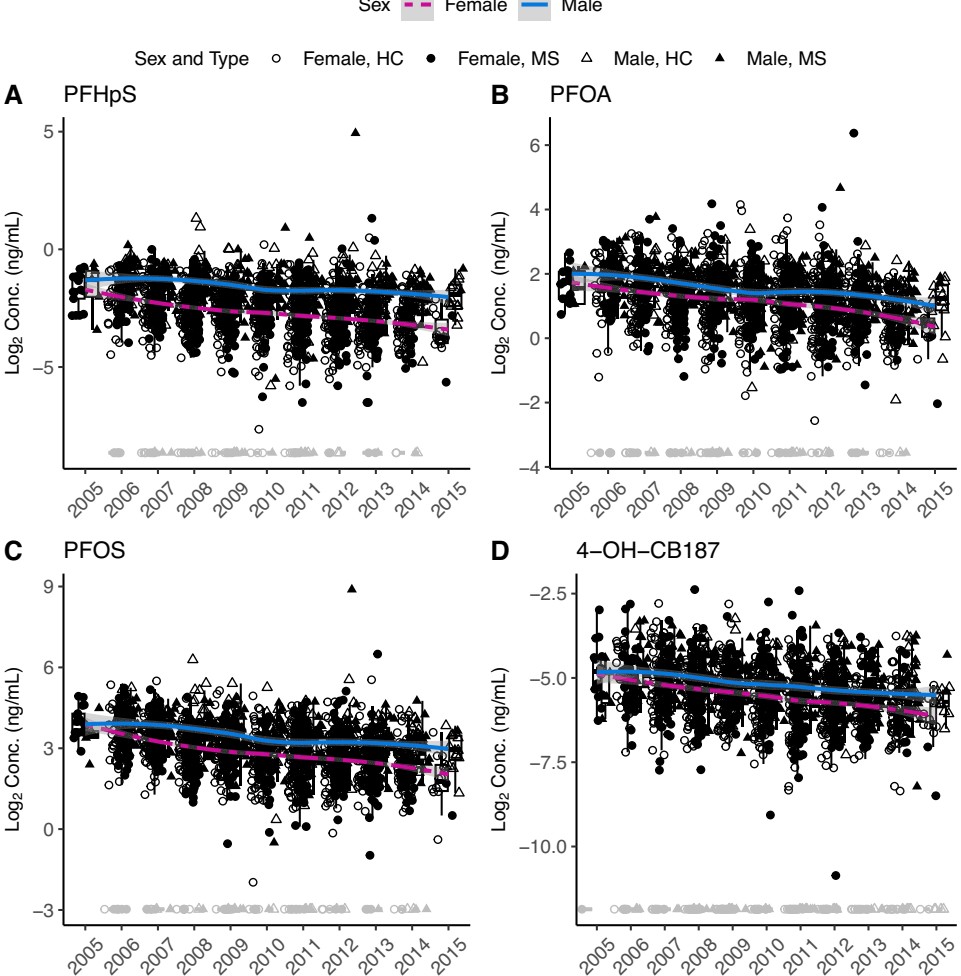

**Fig. 1 | PFAS and OH-PCB concentration versus sample collection year.**
**A** PFHpS, **B** PFOA, **C** PFOS, and **D** 4-OH-CB187 concentration distribution in each sample collection year separated by sex. Plots contain concentration distribution for individuals with MS (filled shape) (n = 907) and HC subjects (empty shape) (n = 907), where males are represented by triangles and females by circles. Imputed concentrations are shown in gray, and measured concentrations in black. The loess regression curve, including 95% confidence bands, has been fitted separately for males (blue solid line) and females (pink dashed line), excluding the imputed concentrations (gray circles and triangles). The upper and lower whiskers indicate 1.5 times the interquartile range from above the upper quartile and below the lower quartile, respectively. Source data are provided as a Source Data file.

children (Supplementary Table S2). These findings illustrate how sex, childbirth, and age affect PFAS and OH-PCB serum concentrations.

## Regional differences in PFAS and OH-PCB concentrations across Sweden

Previous studies have shown that individuals exposed to PFAS-contaminated drinking water over a long period have higher concentrations of PFAS in their blood compared to individuals who only have had exposure from sources other than drinking water[32–34]. Thus, we investigated if regional concentration PFAS and OH-PCB differences could be observed in this cohort. Significant ($P < 0.05$) regional differences were observed in 14 of the 20 environmental contaminants analyzed. At the same time, PFHpS, PFOS, PFNA, perfluoroundecanoic acid (PFUnA), PFDoA, and 4-OH-CB187 did not demonstrate regional differences (Fig. 3, Supplementary Fig. S4-S9). Increased concentrations of PFOA were found in individuals from the Stockholm - Gotland region (Region 3) compared to those from the middle, eastern, and western regions (Regions 2, 4, and 5; $P = 0.035$, $P = 0.012$, and $P = 0.013$). Furthermore, individuals from the northern region (Region 1) had lower PFOA concentrations than those from the eastern and western regions (Region 4 and 5; $P = 0.020$ and $P = 0.022$) (Fig. 3). These findings show significant regional differences in PFAS and OH-PCB concentrations

across Sweden, highlighting the need to consider regional differences when conducting a case-control study involving these compounds.

## Sex-specific concentration differences between MS and Controls

To investigate differences in serum PFAS and OH-PCB concentrations between individuals with MS and HC, differential expression analysis was performed adjusting for age, sex, year of sample collection, BMI at age 20, BMI at sample collection, years of regular smoking, years of passive smoking (indoors), years of irregular smoking, ongoing therapeutic intervention, and number of childbirths. Pairwise comparisons demonstrated significantly decreased PFHpA ($P = 8.05e-3$) concentrations in PMS compared to matched HC subjects. In RRMS, 3-OH-CB153 and 4-OH-CB187 were significantly ($P = 0.045$ and $P = 0.028$) increased compared to matched HC subjects (Fig. 4 and Supplementary Table S3). The regression coefficients and ANOVA $P$ values, including the $P$ value for the interaction term between disease type and sex, are presented in Supplementary Table S4. ANOVA $F$ values and degrees of freedom are presented in Supplementary Table S5. Boxplots of the pairwise comparisons of these PFAS and OH-PCBs and their concentration distributions in the respective sex and disease types are found in Supplementary Fig. S10.

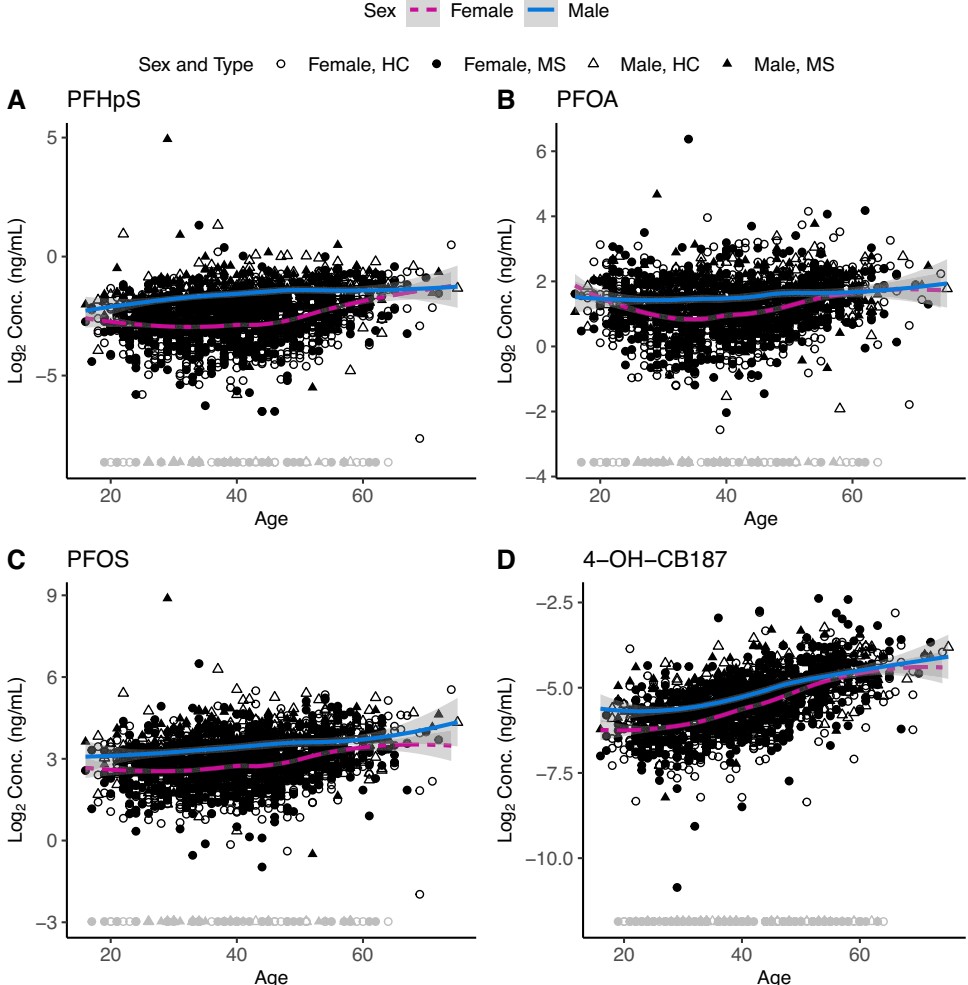

**Fig. 2 | PFAS and OH-PCB concentration versus age. A** PFHpS, **B** PFOA, **C** PFOS, and **D** 4-OH-CB187 concentration versus age in individuals with MS and HC subjects. Plots contain concentration distribution for individuals with MS (filled shape) ($n = 907$) and HC subjects (empty shape) ($n = 907$), where males are represented by triangles and females by circles. Imputed concentrations are shown in gray, and measured concentrations in black. The loess regression curve, including 95% confidence band, has been fitted separately for males (blue solid line) and females (pink dashed line), excluding the imputed concentrations (gray circles and triangles). Source data are provided as a Source Data file.

Due to the significance of sex in the model (Supplementary Tables S4, S5) and the increased prevalence of MS in females, we further investigated possible sex-specific differences in serum PFAS and OH-PCB concentrations between individuals with MS and HC subjects. The pairwise comparison revealed differences between the sexes regarding which PFAS or OH-PCB had significantly altered concentrations between MS and HC (Fig. 4 and Supplementary Table S6). Two short-chain PFAS, PFPeS and PFHpA, were significantly ($P = 6.35e$-$3$ and $P = 1.13e$-$3$) decreased in males with MS compared to their HC, with more pronounced decreased concentrations in PMS and non-significant in RRMS. Further, compared to matched HC subjects, per-fluorodecanesulfonic acid (PFDS) concentrations were significantly ($P = 0.023$) reduced in males with MS, specifically RRMS. In females, however, three long-chain PFAS (PFHxS, PFHpS, and PFOA; $P = 0.033$, $P = 0.020$, and $P = 0.048$) and two OH-PCBs (3-OH-CB153 and 4-OH-CB187; $P = 0.014$ and $P = 6.94e$-$3$) were found to be increased in concentration in MS, specifically PMS, compared to matched HC. In females with RRMS, the concentrations of two long-chain PFAS (PFDS and PFUnA; $P = 0.022$ and $P = 0.034$) and 4-OH-CB187 ($P = 0.047$) were significantly increased compared to HC. Sensitivity analysis demonstrated similar trends when adjusting for Swedish or Nordic-born (Supplementary Tables S7–S14) as well as when the pairing was removed, thus excluding the matching on residential areas

(Supplementary Tables S15–S18). This investigation revealed decreased concentrations of short-chain PFAS in males with MS and increased concentrations of long-chain PFAS and OH-PCB concentrations in females with MS. This demonstrates sex-specific differences in PFAS and OH-PCBs between the MS types compared with HC subjects.

## OH-PCB exposure is associated with an increased MS risk

To explore potential associations between exposure to individual PFAS compounds or OH-PCBs, and the risk of developing MS, logistic regression was used. Short-chain PFAS have been excluded from the risk of developing MS analysis due to their short half-lives. Two models were used, a base model adjusting for age, sex, BMI at sample collection, and BMI at the age of 20, and a fully adjusted model where smoking habits, sun exposure, history of infectious mononucleosis, number of childbirths, Swedish-born or not, and residential area (as a random factor) were included. This study found 4-OH-CB187 and 3-OH-CB153 to be significantly associated with increased odds of MS in the two models (Base model: $P = 0.005$ and $P = 0.026$; Fully adjusted model: $P = 0.010$ and $P = 0.015$) (Table 2 and Supplementary Table S19) regardless of sex. A doubling of serum 4-OH-CB187 concentrations increased the odds of MS by 8.1% (OR = 1.081, 95% CI: 1.019–1.148, $P = 0.010$), while 3-OH-CB153 increased the odds by 5.5% (OR = 1.055, 95% CI: 1.011–1.101, $P = 0.015$). Similar results were also observed when

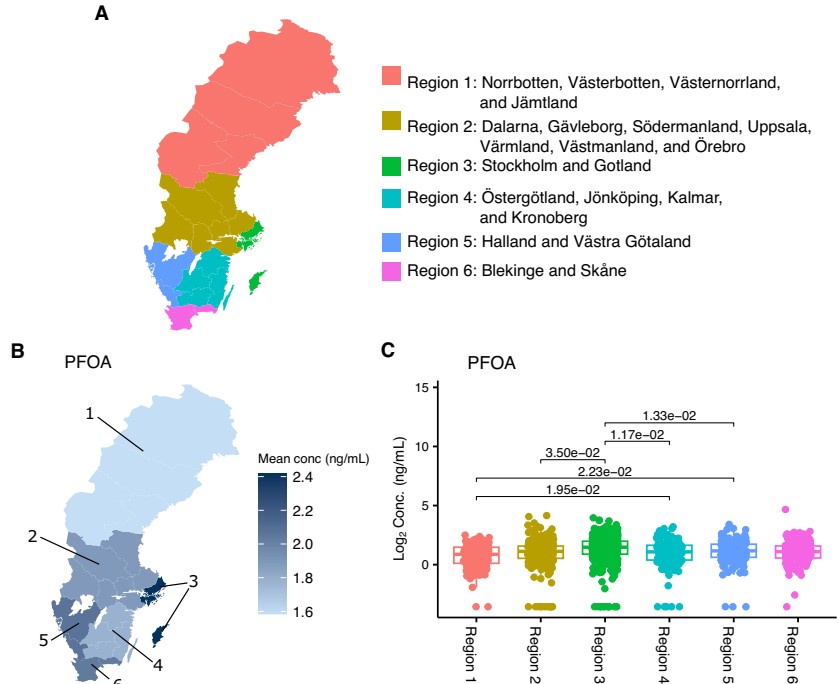

**Fig. 3 | PFOA concentration distribution across Sweden. A** Illustration of the six larger regions in Sweden, based on how healthcare is organized in Sweden. **B** Illustrates the six larger regions' mean concentration (ng/mL) of PFOA. **C** Concentration distribution (log₂ ng/mL) of PFOA and the *P* values for region comparisons. Analysis was made on individuals with MS (*n* = 907) and matched controls (*n* = 907), adjusting for age, sex, sample collection year, BMI at the sample time point, BMI at the age of 20, treatment, years of regular smoking, years of irregular smoking, years of passive smoking (indoor environment), number of childbirths, and disease phenotype. Comparisons were made between all six regions using estimated marginal means based on a linear mixed-effect regression model and *P* values for the two-sided tests were computed using Satterthwaite's degrees of freedom method. Only significant *P* values were included in the boxplot. No adjustments were made for multiple comparisons. Moreover, the upper and lower whiskers indicate 1.5 times the interquartile range from above the upper quartile and below the lower quartile. Source data are provided as a Source Data file.

dividing by sex. The model estimates and ANOVA *P* values for the models are presented in Supplementary Tables S20–S25. A sensitivity analysis was performed, demonstrating similar results when adjusting for Nordic heritage instead of Swedish (Supplementary Tables S26–S29). Further, sensitivity analysis was also performed in relation to the residential area criteria, removing this criteria did not impact the results (Supplementary Tables S30–S33).

## PFAS are associated with a decreased risk of confirmed disability worsening

To investigate if the concentrations of PFAS and OH-PCBs based on compound concentrations as a continuous variable are associated with confirmed disability worsening, we performed a Cox proportional hazard analysis. Here, we defined confirmed disability worsening based on the baseline EDSS as the following: from EDSS 0.0 to 1.0 an increase with ≥1.5 points is required; EDSS 1.5 to 5.0 requires ≥1.0 points increase; and EDSS ≥5.5 requires ≥0.5 points increase[35,36]. Individuals with a registered EDSS in the Swedish MS registry within 3 months of sample collection were included in the analysis, resulting in 551 persons with MS. The general and clinical characteristics of this cohort subset (Supplementary Table S34) were similar to the characteristics of the entire cohort (Table 1). The mean follow-up time for females with RRMS, included in this analysis, was 10.7 ± 3.5 and the median was 10.9 ± 3.5 years. For females with PMS, the mean follow-up time was 9.0 ± 4.1 years and the median was 9.7 ± 4.1 years. The mean and median follow-up times for males with RRMS, included in this analysis, were 10.2 ± 3.7 and 10.5 ± 3.7 years, respectively and 9.0 ± 4.1 and 8.7 ± 4.1 years, respectively for males with PMS. PFOA, PFOS, and PFDA showed a significant (HR = 0.872, 95% CI: 0.766– 0.993, and *P* = 0.039; HR = 0.903, 95% CI: 0.818–0.997, and *P* = 0.043; HR = 0.896, 95% CI: 0.805–0.997, and *P* = 0.044) inverse

association of affecting the risks for confirmed disability worsening, independent of sex (Fig. 5A and Supplementary Table S35).

In males, however, PFOA (HR = 0.795, 95% CI: 0.647–0.997, and *P* = 0.030), PFOS (HR = 0.772, 95% CI: 0.667–0.893, and *P* = 5.08e-4), PFHpS (HR = 0.806, 95% CI: 0.702–0.926, and *P* = 2.23e-3), and PFNA (HR = 0.747, 95% CI: 0.635–0.879, and *P* = 4.33e-4) concentrations also have a significant inverse association with the risk of confirmed disability worsening. Moreover, 3-OH-CB153 demonstrated a positive association with confirmed disability worsening in males, bordering on significance (HR = 1.110, 95% CI: 0.998–1.240, and *P* = 0.055). While in females, we found an inverse association with 4-OH-CB107 and confirmed disability worsening (HR = 0.914, 95% CI: 0.838– 0.996, and *P* = 0.040) (Fig. 3B, C and Supplementary Tables S36, S37). These results indicate that the concentrations of long-chained PFAS are associated with inversed risks of confirmed disability worsening, especially in males. Short-chained PFAS and sulfonamide-containing PFAS (e.g., FOSA, *N*-methylperfluoro-1-octanesulfonamidoacetic acid (N-MeFO-SAA)) are not associated with worsening disability in males or females.

A sensitivity analysis was performed, adjusting the Cox proportional hazard analysis for treatment during the follow-up time. This strengthened PFOA and PFDA inverse association with confirmed disability worsening, while PFOS became borderline significant (HR = 0.906, 95% CI: 0.820–1.000, *P* = 0.053), independent of sex. Moreover, in males PFHpS, PFOS, and PFNA inverse association with confirmed disability worsening was strengthened, while PFOA became non-significant. Further, 3-OH-CB153 demonstrated a positive association with confirmed disability worsening (HR = 1.130, 95% CI: 1.10–1.260, and *P* = 0.027). Finally, in females, the sensitivity analysis strengthened the inverse association between 4-OH-CB107 and confirmed disability worsening. Moreover, PFOA (HR = 0.820, 95% CI:

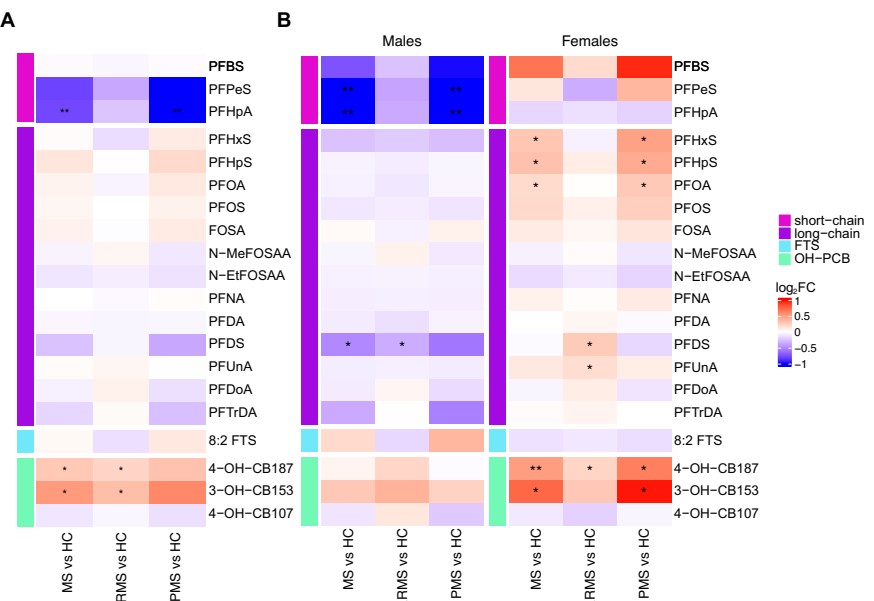

**Fig. 4 | Paired comparison of PFAS and OH-PCBs in individuals with MS and HC subjects. A** Comparisons have been made between the MS group ($n = 907$) and HC subjects ($n = 907$), patients with relapsing-remitting MS (RRMS) ($n = 801$), and matched HC subjects ($n = 801$), and between progressive MS (PMS) ($n = 106$) and matched HC subjects ($n = 106$). PFAS compounds have been grouped into three groups: short-chain PFAS (pink), long-chain PFAS (purple), and FTS (blue), while OH-PCBs (green) are listed separately. **B** Shows the concentration differences between MS and HC subjects when divided by sex. In (**A**, **B**) the intensity of colors represent the log₂ fold change (FC) that are calculated using estimated marginal means based on a linear mixed-effect regression model and $P$ values for the two-sided tests were computed using Satterthwaite's degrees of freedom method. The model was adjusted for age, sex, BMI at the age of 20, BMI at the sample time point, year of sample collection, years of regular smoking, years of irregular smoking, years of indoor passive smoking, number of childbirths, and type of treatment. No adjustments were made for multiple comparisons. Source data are provided as a Source Data file. *$P$ value <0.05; **$P$ value <0.01.

**Table 2 | Associations between single serum PFAS and OH-PCBs, and the odds ratio of MS**

| Substance (ng/ml) | Males and females | | Males | | Females | |
|---|---|---|---|---|---|---|
| | OR (95% CI) | *P* | OR (95% CI) | *P* | OR (95% CI) | *P* |
| PFHxS | 0.971 (0.902–1.047) | 0.444 | 1.004 (0.917–1.099) | 0.933 | 0.990 (0.903–1.085) | 0.827 |
| PFHpS | 1.051 (0.975–1.132) | 0.193 | 1.097 (0.999–1.206) | 0.054 | 1.083 (0.984–1.191) | 0.105 |
| PFOA | 1.050 (0.951–1.158) | 0.334 | 1.111 (0.988–1.248) | 0.078 | 1.089 (0.966–1.227) | 0.162 |
| PFOS | 1.044 (0.967–1.126) | 0.270 | 1.079 (0.982–1.185) | 0.115 | 1.068 (0.971–1.175) | 0.173 |
| FOSA | 1.066 (0.971–1.170) | 0.181 | 1.100 (0.988–1.224) | 0.083 | 1.098 (0.987–1.223) | 0.087 |
| N-MeFOSAA | 1.038 (0.962–1.120) | 0.341 | 1.040 (0.953–1.135) | 0.375 | 1.038 (0.951–1.133) | 0.400 |
| N-EtFOSAA | 1.031 (0.956–1.111) | 0.428 | 1.043 (0.957–1.136) | 0.337 | 1.041 (0.955–1.134) | 0.361 |
| PFNA | 1.021 (0.937–1.113) | 0.637 | 1.066 (0.962–1.182) | 0.224 | 1.053 (0.949–1.169) | 0.330 |
| PFDA | 0.986 (0.909–1.071) | 0.742 | 1.012 (0.921–1.112) | 0.810 | 1.005 (0.915–1.105) | 0.912 |
| PFDS | 1.004 (0.939–1.074) | 0.901 | 1.062 (0.984–1.145) | 0.121 | 1.062 (0.984–1.146) | 0.120 |
| PFUnA | 1.013 (0.942–1.090) | 0.722 | 1.049 (0.960–1.145) | 0.290 | 1.041 (0.953–1.138) | 0.368 |
| PFDoA | 1.017 (0.923–1.120) | 0.738 | 1.035 (0.925–1.158) | 0.550 | 1.026 (0.917–1.149) | 0.651 |
| PFTrDA | 0.994 (0.946–1.045) | 0.817 | 1.008 (0.951–1.068) | 0.794 | 1.007 (0.950–1.067) | 0.821 |
| 8:2 FTS | 1.028 (0.977–1.082) | 0.293 | 1.042 (0.983–1.105) | 0.166 | 1.040 (0.980–1.102) | 0.194 |
| 4-OH-CB187 | 1.081 (1.019–1.148) | 0.010 | 1.092 (1.018–1.170) | 0.013 | 1.091 (1.017–1.169) | 0.015 |
| 3-OH-CB153 | 1.055 (1.011–1.101) | 0.015 | 1.056 (1.005–1.110) | 0.031 | 1.053 (1.002–1.107) | 0.043 |
| 4-OH-CB107 | 0.989 (0.937–1.043) | 0.674 | 0.979 (0.921–1.041) | 0.503 | 0.976 (0.918–1.038) | 0.437 |

Odds ratios are calculated per doubling of substance serum concentrations using logistic regression. Calculations based on the fully adjusted model. No corrections were made for multiple comparisons. Adjusted for age, sex, BM at sample collection, BMI at the age of 20, number of childbirths, regular smoking, irregular smoking, history of infectious mononucleosis, sun exposure, Swedish-born, and residential area (as a random effect).

0.695–0.967, and $P = 0.019$) and PFDA (HR = 0.855, 95% CI: 0.750–0.975, and $P = 0.019$) also showed an inverse association with confirmed disability worsening (Supplementary Tables S38–S40).

**PFAS and albumin, the possibility of reverse causation**

Albumin has been shown to be the major PFAS carrier in human blood[37,38]. Moreover, a meta-analysis demonstrated lower concentrations of serum albumin in people with MS compared to HC subjects[39]. This could potentially present a case of reverse causation. Thus, we further investigated if serum albumin concentrations affected the risk of confirmed disability worsening using Cox proportional hazard analysis. The results demonstrated no association between serum albumin concentrations and risk of confirmed disability worsening (Supplementary Table S41). Moreover, when incorporating

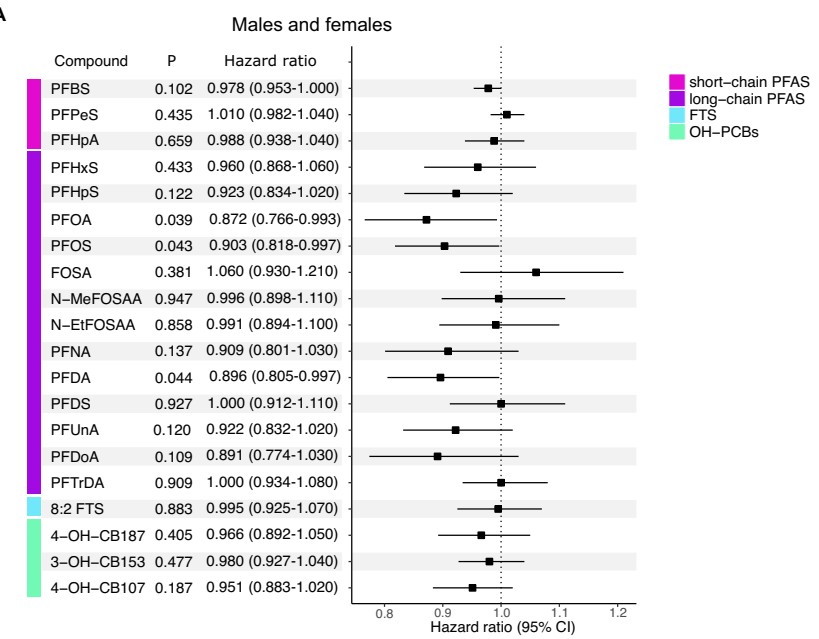

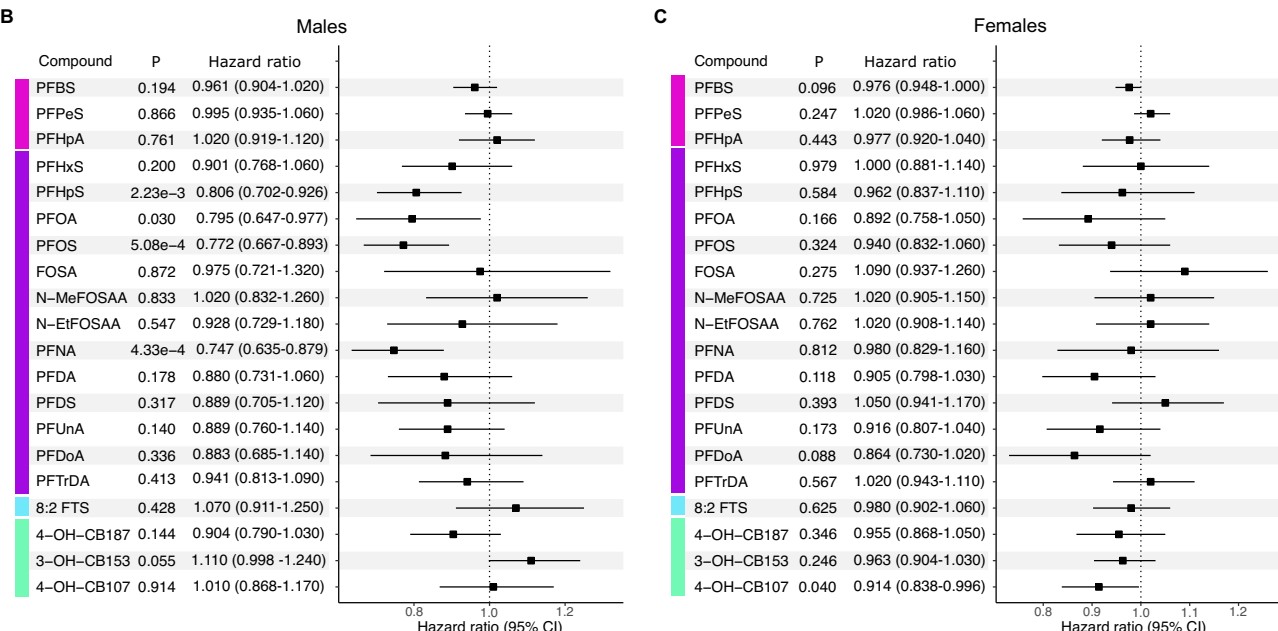

**Fig. 5 | Confirmed disability worsening.** Hazard ratios, confidence intervals (95% CI), and *P* values for PFAS and OH-PCB concentrations associated with confirmed disability worsening were estimated using Cox proportional hazard analysis. No adjustments were made for multiple comparisons. The hazard ratio value is represented by the square and error bars represent its 95% CI. **A** Including males and females (*n* = 551), **B** only males (*n* = 137), and **C** only females (*n* = 414). PFAS compounds have been grouped into three groups: short-chain PFAS (pink), long-chain PFAS (purple), and FTS (blue), while OH-PCBs (green) are listed separately. Source data are provided as a Source Data file.

serum albumin concentrations in the model evaluating PFAS and OH-PCBs association with confirmed disability worsening, the inverse associations remained. Further, 3-OH-CB153 showed a significant positive association with confirmed disability worsening in males only (HR = 1.130, 95% CI: 1.010–1.260, and *P* = 0.035) (Supplementary Tables S42–S44), thus indicating that the inverse associations observed are not due to reverse causation.

**No impact on the risk of transitioning from RRMS to SPMS**
We also investigated if the concentrations of PFAS and OH-PCBs at sample collection affect the risk of transitioning from RRMS to SPMS. A borderline inversed risk was associated with the

concentration of PFDoA (HR = 0.813, 95% CI: 0.660–1.000, and *P* = 0.051). Apart from that, neither PFAS nor OH-PCB concentrations affected the time to transition from RRMS to SPMS (Fig. 6 and Supplementary Table S45). Due to the low number of transition events (females *n* = 59, and males *n* = 30), no further analysis on the effect of sex was made.

Sensitivity analysis where treatment during the follow-up time was included increased the *P* value of PFDoA, however similar trends were still observed (Supplementary Table S46). Moreover, when adjusting for serum albumin concentrations PFDoA demonstrated a significant inverse association with risk of transitioning from RRMS to SPMS (HR = 0.809, 95% CI: 0.655–0.999, and *P* = 0.048) (Supplementary Table S47).

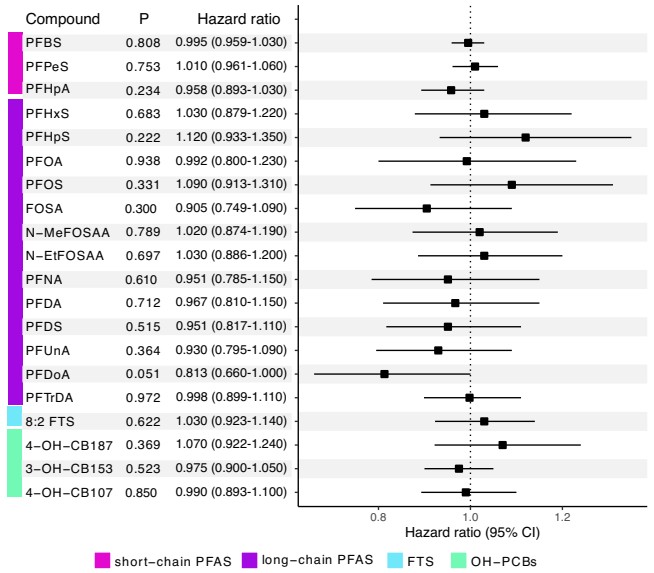

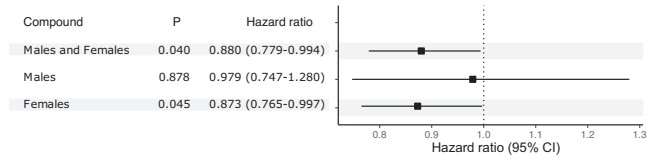

**Fig. 6 | RRMS to SPMS transition.** Hazard ratios, confidence intervals (95% CI), and P values for concentrations associated with a transition from RRMS to SPMS (n = 801) were estimated using Cox proportional hazard analysis. No adjustments were made for multiple comparisons. The hazard ratio value is represented by the square and error bars represent its 95% CI. Important to note is that hazard ratios are calculated on $\log_2$ concentrations. PFAS compounds have been grouped into three groups: short-chain PFAS (pink), long-chain PFAS (purple), and FTS (blue), while OH-PCBs (green) are listed separately. Source data are provided as a Source Data file.

**Fig. 7 | 25(OH)D3 effect on confirmed disability worsening.** Hazard ratios, confidence intervals (95% CI), and P values for 25(OH)D3 levels associated with confirmed disability worsening were estimated using Cox proportional hazard analysis in males and females (n = 515) together and in males (n = 137) and females (n = 414) separately. The hazard ratio value is represented by the square and error bars represent its 95% CI. Source data are provided as a Source Data file.

## PFAS levels correlate with vitamin D3 levels

In silico and in vitro studies have recently demonstrated evidence of PFOA binding to the vitamin D receptor and altering vitamin D-responsive gene activity[40]. Additionally, the vitamin D receptor has been identified as a potential target for the toxic effects of PFAS[41]. Thus, due to vitamin D's inverse association with the risk of developing MS[7,8], we investigated whether these compounds correlated with 25-hydroxyvitamin D3 (25(OH)D3) levels and if disease-specific correlations could be observed.

To investigate if MS and HC differ in how 25(OH)D3 is associated with PFAS and OH-PCBs, Spearmans' rank correlation analyses, regardless of disease, as well as separated on disease, were made. Since we had observed significantly (P = 0.034) higher 25(OH)D3 levels in females as well as seasonal changes in 25(OH)D3 levels depending on when the blood sample was collected regardless of disease presence, the correlations were adjusted for these covariates. In addition, due to the sex-specific differences observed in previous analyses regarding PFAS and OH-PCB concentrations, correlations to 25(OH)D3 were also analyzed for males and females separately. Sex-specific correlations were adjusted for blood sample collection

month only. Moreover, correlations in females were also adjusted for number of childbirths.

In general, PFNA and PFDA had significant positive correlations with 25(OH)D3 serum levels (r = 0.142 and P = 1.09e-8; r = 0.088 and P = 4.35e-4), which is in line with previous observations[42]. Moreover, PFPeS, 8:2 FTS, 4-OH-CB187 and 4-OH-CB107 had significant negative correlations with 25(OH)D3 serum levels (r = −0.235 and P = 1.36e-21; r = −0.161 and P = 8.55e-11; r = −0.138 and P = 2.85e-8; r = −0.123 and P = 8.43e-7). The correlations, in general, were stronger in females compared to male subjects (Supplementary Table S48).

## PFAS, vitamin D, and risk of confirmed disability worsening

To further analyze if 25(OH)D3 serum levels affected the risk for confirmed disability worsening, we performed a Cox proportional hazard analysis. An inverse association of 25(OH)D3 with confirmed disability worsening was observed independently of sex and in females only (Fig. 7 and Supplementary Table S49). However, this observed association might be attributed to reverse causation, where disability or behavioral changes result in reduced sun exposure. Due to this inverse association and PFAS correlation with 25(OH)D3 levels, we investigated whether PFAS inverse association with confirmed disability worsening was associated with the individuals' 25(OH)D3 levels. When incorporating 25(OH)D3 levels in the Cox proportional hazard model for PFAS, the inverse association with confirmed disability worsening was strengthened. The inverse association of PFOA, PFOS, and PFDA with confirmed disability worsening remained, with overall lower P values. In addition, PFBS, PFNA, and PFDoA also exhibited significant (P = 0.039, P = 0.047, and P = 0.023) inverse associations. Stratifying by sex had borderline effects compared to not adjusting for 25(OH)D3 levels; PFDoA now also displayed inverse significant associations in females (HR = 0.811, 95% CI: 0.685–0.959, and P = 0.014), and 3-OH-CB153 demonstrated significant associations in males (HR = 1.142, 95% CI: 1.009 –1.292, and P = 0.036) (Supplementary Fig. S11 and Tables S50–S52). Adding 25(OH)D3 levels in the Cox proportional hazard model for the time to transition from RRMS to SPMS did not influence the results (Supplementary Fig. S12 and Supplementary Tables S53, S54). These results indicate that the PFAS inverse association with confirmed disability worsening cannot be fully explained by the levels of 25(OH)D3.

## Discussion

The everyday use of PFAS, their persistent occurrence, and bioaccumulative and toxic properties have made PFAS a public health concern. Associations between PFAS and liver toxicity, developmental toxicity, certain types of cancer, endocrine disruption, and immunosuppressive effects have been reported and reviewed in humans[14,19–21,43]. In this study, we investigated the effects of PFAS and OH-PCB serum concentrations on the risk of developing MS and the disease progression, associations to 25(OH)D3, and potential concentration differences between individuals with MS and HC subjects. We found a significant association between two OH-PCBs and the risk of developing MS. Moreover, we found significant inverse associations between several PFAS compounds and confirmed disability worsening in MS. Finally, we also found apparent sex-specific differences in PFAS and OH-PCB concentrations between individuals with MS and HC subjects. In our case-control study, males diagnosed with MS had lower concentrations of short-chain PFAS compounds compared to their matched controls, whereas females with MS displayed higher concentrations of long-chained PFAS compounds compared to their matched controls. These differences were more pronounced in both males and females when comparing those with progressive MS to controls.

A growing body of evidence points to an increase in the prevalence and incidence of autoimmune diseases in the general population[44]. Endocrine-disrupting chemicals, including PFAS, have emerged as one group of potential risk factors in autoimmune

diseases[14]. PFAS have been reported to modulate the immune system and cause human immunosuppression. The most convincing evidence of immunosuppression due to PFAS exposure in humans is in studies demonstrating decreased antibody levels following vaccination in children exposed to PFOA and PFOS[14,19–21]. Moreover, in rodents, a dose-responsive suppression of the T cell-dependent antibody response, an assay focusing on the humoral arm of the adaptive immune system, has been associated with PFOA and PFOS exposure[45,46]. In vitro studies provide further evidence for PFAS immunomodulating effects through modulation of oxidative stress, $Ca^{2+}$-signaling, nuclear receptor, and cytokine levels (reviewed 2023)[14]. However, previous knowledge of their role in MS is scarce, and findings on the role of endocrine-disrupting chemicals across autoimmune diseases are, at large, inconsistent (reviewed in 2023)[14]. This is likely associated with the fact that there are more than 120 different autoimmune diseases, of which some are rare, with very few cases. Additionally, most studies are limited to animal, single exposure, or cross-sectional studies, with few longitudinal studies on humans.

Previous studies have found inverse associations between exposure to endocrine-disrupting chemicals and the incidence of autoimmune diseases. Inverse associations between the incidence of type 1 diabetes (T1D) with exposure to PFAS (PFNA, PFHxS, PFOA, and PFOS) have been reported, affecting both sexes equally[47]. Diabetes and albuminuria, an early sign of kidney damage in diabetes, have been associated with lower perfluoroalkyl acid levels[48]. It has been suggested that albuminuria drives an enhanced excretion of perfluoroalkyl acids due to kidney function decline[37,48], thus posing a risk of reverse causation in terms of PFAS association with the risk of developing T1D. More recently, the incidence of rheumatoid arthritis (RA) was also found to have an inverse association with PFAS exposure (PFHxS, PFOA, PFNA, and PFDA). The association was significant only in females[28]. Furthermore, PFOA exposure has also been associated with an increased risk of developing ulcerative colitis[27].

Exposure to PCBs has previously been associated with RA only in females[49]. Moreover, a long-term follow-up study demonstrated that exposure to PCBs during a long time period significantly increased mortality in females with systemic lupus erythematosus (SLE). However, the mechanism by which PCBs affect the development of SLE has not been studied[50,51]. In T1D, an inverse association between PCB153 exposure and T1D incidences has been reported[52]. Notably, PCBs association with MS has not been studied before. In this study, we found the incidence of MS to be positively associated with exposure to two OH-PCBs (4-OH-CB187 and 3-OH-CB153), independent of sex or native-born status. Studies have shown that exposure to PCBs can cause oxidative stress in rats, specifically in the cerebral cortex, cerebellum, and hippocampus, as well as in nerve cell cultures[53,54]. Moreover, PCB28-induced concentration-dependent cell death in various neurons has been demonstrated in vivo[55]. These effects could be one potential explanation for OH-PCBs association with an increased risk of developing MS, however, further studies are needed to understand the mechanism by which exposure to OH-PCBs impacts the risk of developing MS.

Moreover, in terms of disability worsening, we found an inverse association between PFOA, PFOS, and PFDA exposure and confirmed disability worsening. Furthermore, when only considering males, we saw a pronounced inverse association between PFOS, PFOA, PFHpS, and PFNA and confirmed disability worsening. Moreover, 3-OH-CB153 demonstrated a borderline-significant association with increased risk for confirmed disability worsening in males. In women, however, an inverse association with 4-OH-CB107 was found. These inverse associations concerning disease progression and worsening have not been observed in MS or other autoimmune diseases. The findings made herein suggest that PFAS may suppress the immune system in MS, resulting in a slower disease progression and lower risk of disability worsening.

Increased serum PCB153 levels have been associated with decreased serum testosterone levels in males[56]. Moreover, lower testosterone levels have been associated with disability in males with MS[57]. In animal models, testosterone has been observed to have beneficial effects on both inflammatory and neuroprotective mechanisms (reviewed in 2018)[58]. This could potentially explain the positive association between 3-OH-CB153 and the risk for confirmed disability worsening observed in males.

A meta-analysis study has previously demonstrated decreased serum albumin concentrations in people with MS compared to HC subjects[39]. Moreover, a recent study demonstrated that higher serum albumin concentrations are associated with a reduced risk of MS[59]. Due to PFAS protein-binding properties, albumin has been shown to be a major PFAS carrier in blood[37,38,59]. Thus, we further investigated if this could potentially present a cause of reverse causation in terms of PFAS inverse association with confirmed disability worsening. Adjusting for serum albumin concentrations did not impact the inverse associations observed between PFAS compounds and confirmed disability worsening, thus indicating that observed associations were not due to reverse causation.

Low levels of vitamin D have been associated with an increased risk of developing MS, however, some uncertainty remains[7,8]. Further, vitamin D levels have been associated with disease activity in confirmed MS, specifically MRI activity[60] and No Evidence of Disease Activity-3 status[61]. However, clinical trials of vitamin D supplementation have not been conclusive, and there is not yet sufficient evidence to support vitamin D therapy in MS[62]. Although associations between PFAS exposures and vitamin D biomarkers have also been observed previously[63,64], the underlying biological mechanism explaining these associations remains unclear. Based on the aforementioned observations of competitive binding of PFOA to the vitamin D receptor with subsequent inhibition of vitamin D-responsive genes[40], the associations between PFAS exposure and vitamin D may reflect this antagonistic role. While this interaction with the vitamin D receptor has only been shown in vitro for PFOA, an in silico study predicted that many commercially important PFAS compounds would bind vitamin D receptors equally strong or stronger than PFOA, implying that many PFAS could exhibit the same function[41]. This is especially concerning since humans are exposed to complex mixtures of PFAS, which may amplify the effect on the vitamin D receptor. Therefore, we further investigated if PFAS association with vitamin D3 (specifically 25(OH)D3) could potentially explain the inverse association observed between PFAS compounds and confirmed disability worsening in males with MS. When adjusting for 25(OH)D3 levels, the previously observed inverse associations were in fact strengthened, and also now PFBS, PFNA, and PFDoA demonstrated inverse association as well, thus suggesting that the inverse effects by PFAS on the disease course cannot be fully explained by presumed protective 25(OH)D3 levels.

Moreover, prior research has established a sex difference in MS, showing that approximately three times more females than males are affected by this disease[65,66]. This sex difference is more pronounced in RRMS than in PMS. In PPMS, the female-to-male ratio is almost even[67]. Even though males have a lower risk of developing MS, it has been suggested that the male sex is a risk factor for a progressive onset. The current data shows a clinical follow-up period of eight to 18 years. The male sex has also been associated with a shorter time to disability landmarks regarding EDSS and progression[68,69], which could potentially explain why PFAS association to disease progression was mainly observed in male MS subjects.

Previous findings of PFAS in association with MS are scarce. A single Danish case-control study revealed decreased PFOS concentrations in MS compared to HC subjects, speculating on the potential immunosuppressive effects of PFAS[29]. Our study observed significant differences in PFAS and OH-PCB concentration between individuals

with MS and HC subjects. However, in contrast to the Danish study, we found elevated concentrations of long-chain PFAS (PFHxS, PFHpS, and PFOA) in females with MS, while concentrations of short-chain PFAS (PFPeS and PFHpA) were decreased in males with MS compared to matched HC subjects. These differences were more prominent in PMS, an MS subtype not considered in the Danish study.

Consistent with findings from prior epidemiological studies[70–72], we observed significant differences in PFAS and OH-PCB concentrations between males and females independently of disease presence. Moreover, we showed significant regional differences across Sweden. Increasing concentrations with age were also observed, which is associated with the compound's bioaccumulative properties. Finally, we also observed decreasing compound concentrations with the sample collection year, reflecting these compounds' phasing out. We also observed strong correlations between the concentrations of these substances, especially within short- and long-chained PFASs, sulfonamide-containing PFAS, and OH-PCBs (Supplementary Figs. S13, S14 and Tables S55, S56). This demonstrated the value of matching between cases and HC subjects concerning sex, age, sample collection year, and residential area when conducting a case-control study involving PFAS and OH-PCB compounds.

The data used in this study (Epidemiological Investigation of Multiple Sclerosis (EIMS) cohort) was initially designed to evaluate environmental factors' effect on MS. The reliance on a single-time serum sample to infer the long-term effects of PFAS exposure is a limitation. Another limitation is the lack of data regarding body fat, some PFAS have demonstrated an inverse association with body fat measurements specifically in women[61,73]. Another limitation is that no data is available regarding the number of menstruations. This could act as a confounder, given that the protein-binding properties of PFAS in the blood during menstrual bleeding constitute a substantial source of PFAS excretion[70].

Moreover, previous research has demonstrated that PFAS can be transferred to the fetus during pregnancy and excreted in breast milk, leading to a substantial maternal offload of PFAS during pregnancy and breastfeeding periods[22,74]. We demonstrated generally lower PFAS concentrations in females compared to males. Moreover, females with biological children had lower PFAS concentrations compared to females without biological children, no such difference was observed in males. These results reflect the female-specific PFAS excretion routes associated with pregnancy and breastfeeding, potentially explain the sex-specific results observed. Regardless, adjusting for the number of childbirths in the model may not fully adjust for the female-specific PFAS and OH-PCBs excretion routes. We can thus not exclude the possibility that the difference in childbirth between HC (71.7%) and MS (62.4%) may, to some extent, contribute to observed differences. However, among those who have given birth, there is no significant difference in the number of childbirths between HC (mean 2.1) and MS (mean 2.0). Another is the potential risk of false positives arising from multiple comparisons. We see strong associations between PFAS and PCBs, violating adjustment for correcting for multiple comparisons, given they are assumed to pose similar effects in vivo. Finally, the exclusive inclusion of participants from Sweden in this study limits the generalizability of our findings and conclusions to other geographic areas. Despite these limitations, the study provides valuable insights into PFAS and OH-PCB associations with MS, disease progression, and the role of 25(OH)D3 in this context.

In conclusion, we demonstrate a significant association between 4-OH-CB187 and 3-OH-CB153 exposure and increased risk of developing MS. Moreover, we also demonstrated significant inverse associations between concentrations of PFOA, PFOS, and PFDA exposure and confirmed disability worsening in both males and females. In addition, PFHpS and PFNA showed a significant inverse association with confirmed disability worsening in males. Interestingly, for 3-OH-CB153 a non-significant positive association with risk for confirmed disability

worsening was observed in males with MS. Adjusting for presumable protective effects of 25(OH)D3 serum levels strengthened the inverse associations found for the long-chained PFOA, PFOS, PFNA, PFDA, and PFDoA and now also included the short-chained PFBS. Moreover, adjusting for serum albumin concentrations did not impact the inverse associations observed, indicating that the results are not due to reverse causation. These findings add to the research on OH-PCBs and PFAS in autoimmune disease, now in MS.

## Methods

### Ethical statement
The sample collection and storage were made according to local ethical requirements and applicable national laws concerning human population genetic studies. The relevant ethical permissions have the following numbers; EPN Stockholm: 04-252/1-4 and 2017/2386-32; from the Swedish Ethical Review Authority 2021-00702. All participants provided written informed consent.

### Cohort
This study is based on the Swedish population-based case-control study EIMS[75]. Incident cases of MS were recruited from neurological clinics throughout Sweden, including all Swedish university hospitals, between 2005 and 2015. All cases fulfilled the McDonald criteria[76,77]. For each case, controls were randomly selected from the national population registry and matched to the case's age, sex, and residential area. All participants answered a detailed questionnaire regarding environmental exposure and lifestyle factors. The MS cases were matched to the Swedish National MS registry to extract data related to disease course, expanded disability status scale (EDSS), disease-modifying treatment, and if conversion from RRMS to SPMS occurred[78]. Based on the full questionnaire data available, serum was collected for the newly diagnosed MS person and a fully matched control (age, sex, year of collection, and living area), and the MS person was included in the Swedish National MS registry, 907 MS subjects and 907 matched HC subjects were analyzed in the current study.

### PFAS and OH-PCBs analysis
In total 1814 serum samples were analyzed. The serum samples were randomized in batches of 32 samples per batch, each batch also included two water blanks, one with internal standards and one without, and one external quality control sample (QCE, large pooled serum sample). The samples were prepared batchwise accordingly, serum samples (50 μL) were thawed, followed by the addition of internal standards (50 μL) and ice-cold methanol (100 μL) for protein precipitation at −20 °C for 60 min. The samples were vortexed and centrifuged for 15 min at 21,100 RCF and 4 °C, and 100 μL of the supernatant was transferred to an LC vial. A pooled QC sample for all batches (QCBP) was created by aliquoting 10 μL from each sample. Calibrators ($n = 7$) and calibration QC samples ($n = 3$) were prepared in newborn bovine serum and spiked to seven concentrations within the 0.02–80 ng/mL range for all compounds where native and isotope-labeled standards were available. Where only isotope-labeled standards were available, one-point calibration compared with the corresponding internal standard was performed (Supplementary Table S1). During sample analysis, the samples were analyzed batchwise, blanks and QCE was analyzed at the beginning and end of each batch and the QCBP was analyzed every eight injection throughout the analysis. Calibration curves and calibration QCs were analyzed at the beginning and end of each weekly run. For the sample analysis, a reversed-phase liquid chromatography column (Accucore C18, 100 × 2.1 mm, 2.6 μm, Thermo Scientific) using an Ultimate 3000 liquid chromatography system (Thermo Scientific) connected to a high-resolution hybrid quadrupole Q Exactive Orbitrap mass spectrometer (Thermo Scientific). The mobile phases consisted of 10 mM ammonium acetate and 0.1% acetic acid in water (mobile phase A)

and methanol (mobile phase B). The elution gradient was as follows: 5% B for 1 min, 5–100% B over 9.5 min, 100% B for 3 min, followed by re-equilibration at 5% B for 4 min, for a total analysis time of 17.5 min. The flow rate was 0.6 mL/min throughout, and the column temperature was 55 °C. The orbitrap mass spectrometer was operated in negative ionization mode at a resolution of 70,000 in the m/z range 212.5–750. The source parameters were as follows, the capillary temperature was 275 °C, the spray voltage was 2.5 kV, and the auxiliary gas heater was 450 °C, and the sheath gas, sweep gas and auxiliary gas flows were 55, 3, and 15, respectively. All data were processed using TraceFinder 4.1 (Thermo Scientific) and exported for further analysis.

PFAS can be divided into long- and short-chain PFAS depending on the number of carbons. Within perfluoroalkyl sulfonic acids, long-chain PFAS contains six or more carbons, and short-chain contains less than six. In perfluoroalkyl carboxylic acids, however, long-chain PFAS contains seven or more fluorinated carbons, and short-chain has less than seven[79,80]. This classification will be used henceforward.

## Vitamin D analysis
A total of 1814 serum samples were analyzed. The samples were randomized into batches, each containing 32 serum samples. Additionally, each batch included two water blanks–one with internal standards and one without–as well as an external quality control sample (QCE). Briefly, serum samples (50 μL) were thawed and precipitated by adding ice-cold methanol (150 μL), including internal standards. The samples were vortexed and centrifuged, and 100 μL of the supernatant was transferred to an LC vial. A pooled QC sample for all batches (QCBP) was created by aliquoting 10 μL from each sample. During sample analysis, the samples were analyzed in batches. Blanks and QCE were analyzed at the beginning and end of each batch, while the QCBP was analyzed after every eighth injection throughout the run. Sample analysis was performed using an Ultimate 3000 LC system and a high-resolution hybrid quadrupole Q Exactive Orbitrap mass spectrometer (Thermo Scientific). The samples were injected (2 μL) onto an Accuore aQ RP C18 column (100 × 2.1 mm, 2.6 μm, Thermo Scientific). The mobile phases were water +0.1% formic acid (mobile phase A) and 90% methanol, 10% isopropanol, and 0.1% formic acid (mobile phase B). The elution gradient was as follows: 0% B for 3 min, 0–10% B over 2.5 min, 10–100% B over 8.5 min, followed by 4 min at 100% and re-equilibration at 0% B for 4 min for an analysis time of 22 min. The column temperature was set to 55 °C. The mass spectrometer was operated in positive ionization mode at resolution 70,000 in the m/z range of 70–900 during the first 5 min and 148–900 m/z in the following 15 min. The source parameters were set as follows, the capillary temperature was 320 °C, the spray voltage was 3.5 kV, and the auxiliary gas heater was 450 °C. The sheath gas, sweep gas, and auxiliary gas flows were 55, 3, and 15, respectively. 25-hydroxyvitamin D3 (25(OH)D3) was identified from an authentic standard. However, no calibration standards or internal standards were used for quantitation, and peak areas were used as the quantitative measure. All data was processed using TraceFinder 4.1 (Thermo Scientific) and exported for further analysis.

## Serum albumin analysis
Albumin was measured with bromocresol purple reagents from Abbott Laboratories (Reagent kit 04U45) Abbott Park, Il, USA) on a BS-430 chemistry analyzer (Mindray, Shenzhen, China). The method is based on the binding of bromocresol purple with human albumin. The method is linear in the measuring interval 3–90 g/L. The lower limit of quantification for the method was 3 g/L. The instrument settings used were endpoint measurement with primary wavelength at 605 nm, secondary wavelength at 660 nm, 90 uL R1, and a sample volume of 2 uL. The total method coefficient of variation was 1.5% at 38 g/L.

## Definition of sun exposure habit
To determine sun exposure habits, during the 5-year period prior to study inclusion, participants answered three questions regarding their sun exposure habits. Each question could be answered on a four-point scale. An index was constructed by summing the scores, resulting in a range from three (the lowest exposure) to 12 (the highest exposure. Finally, sun exposure habits were dichotomized based on the 25th percentile in HC.

## Statistical analysis
R version 4.2.3 was used for all computations[81]. Since PFAS and OH-PCB concentrations were not normally distributed, concentrations were log$_2$-transformed. Compounds not detected in >20% of the studied persons were excluded from further analysis. Values below the detection limit were imputed with half of the lowest measured concentration for each compound. The same approach was used for relative intensities.

## Differential expression analysis
For the differential expression analysis, a linear mixed-effect model was used, and the covariates considered included age, year of sample collection, sex, body mass index (BMI, calculated as kg/m$^2$) at sample collection, and BMI at the age of 20. For individuals between 15 and 25 years of age, where either BMI at sample collection or BMI at the age of 20 was missing, it was assumed that BMI at sample collection and BMI at the age of 20 were the same. Furthermore, lifestyle factors that could influence compound concentrations were included in the base model, and the ANOVA P value was assessed. The following lifestyle factors were evaluated: years of regular smoking, years of irregular smoking, years of indoor passive smoking, years of Swedish snuff consumption, volume of 40% alcohol (cl) consumed the week before sample collection, infectious mononucleosis, number of childbirths, and type of ongoing treatment. Types of treatment were divided into the first line, second line, other-, and no treatment (Supplementary Table S57). The final model included the following covariates: age, sex (male as reference), BMI at the age of 20, BMI at sample time point, year of sample collection, years of regular smoking, years of irregular smoking, years of indoor passive smoking, number of childbirths, and type of treatment. To investigate potential sex differences further, an interaction term with sex and disease type was included in the model. Three sensitivity analyses were performed, controlling for Swedish-born status and Nordic-born status, as well as excluding the pairing in residential areas.

Disease courses were divided into two main types: RRMS and progressive MS (PMS), including SPMS and PPMS. Even though SPMS and PPMS present differently, they are part of the same disease spectrum[81]. In addition, there are no quantitative differences between these two phenotypes of progressive MS regarding immunopathology, disease activity, and lesion morphology. Finally, genome-wide association studies have not found genetic variants differentiating these two phenotypes[82], thus supporting this general grouping of PMS.

## Logistic regression
To evaluate individual compounds concentrations association with the odds of MS, logistic regression was used. Two models were employed: a base model and a fully adjusted model. The base model was adjusted for age, sex, BMI, and BMI at the age of 20. Further, lifestyle factors were included in the base model, and the ANOVA p value was assessed. The following lifestyle factors were evaluated: regular smoking, irregular smoking, passive smoking (indoors), Swedish snuff use, history of infectious mononucleosis, number of childbirths, sun habits, Swedish or Nordic-born status. Lifestyle factors with an ANOVA p value below 0.01 were included in the fully adjusted model. Thus, the covariates in the fully adjusted model were: age, sex, BMI, BMI at the age of 20, number of childbirths, regular smoking, irregular smoking, history

of infectious mononucleosis, sun habits, born in Sweden or not, and residential area treated as a random effect. Analysis were also performed stratified by sex. Two sensitivity analysis were performed, the first excluding residential area as a random effect, and the second where Nordic-born status was evaluated.

### Cox proportional hazard analysis

To evaluate if the concentrations of PFAS and OH-PCB at baseline have an impact on the outcome in terms of confirmed disability worsening based on EDSS measurements, a Cox proportional hazard model was used, adjusting for the following covariates: age, sex, time difference between diagnosis and sample time point, EDSS at time of sample collection, current BMI, treatment, years of regular smoking, years of irregular smoking, years of indoor passive smoking, number of childbirths, and disease phenotype. The time variable in these analyses was months from blood sample collection to confirmed disability worsening occurrence or the entire follow-up period. In addition, analyses were performed with another Cox proportional hazard model, which included the same covariates as the first model, as well as 25(OH)D3 levels and the month of blood sample collection. Individuals who did not have a registered EDSS within three months of sample collection were excluded, leaving 551 patients (414 females and 137 males) for final analysis. In addition, the confirmed disability worsening analysis was also divided by sex. To ensure that the worsening of EDSS was not a relapse, the increase in EDSS needed to remain for at least three months to be classified as worsening. Confirmed disability worsening was defined as the following increases in EDSS: from EDSS 0.0 to 1.0, $\geq$1.5 points; from EDSS 1.5 to 5.0, $\geq$1.0 points; and from EDSS $\geq$5.5 $\geq$ 0.5 points[35,36]. Finally, the follow-up time for this cohort is up to 18 years.

We also analyzed if the concentrations of PFAS and OH-PCB at baseline affected the time to transition from RRMS to SPMS. In this analysis, only patients with the RRMS diagnosis at the initial sample time point were included, and the Swedish MS registry was used to identify the time of transition to SPMS. Here, the Cox proportional hazard models included the following covariates: age, sex, time difference between diagnosis and sample time point, current BMI, treatment, years of regular smoking, years of irregular smoking, years of indoor passive smoking, and number of childbirths.

### Serum albumin and Cox proportional hazard analysis

As albumin is a major carrier of PFAS in human blood, correlations have been observed between PFAS-related compounds and serum albumin[37,38]. Moreover, lower concentrations of serum albumin have been observed in people with MS compared to HC subjects[39]. Thus we further analyzed if serum albumin concentrations could potentially explain the inverse associations observed between PFAS and confirmed disability worsening. Firstly, the effect of albumin on the risk for confirmed disability worsening was assessed using Cox proportional hazard analysis. The following covariates were adjusted for: age, sex, the time difference between diagnosis and sample time point, EDSS at the time of sample collection, BMI at sample time point, years of regular smoking, years of irregular smoking, years of indoor passive smoking, and treatment. Albumin concentrations were also added to the Cox proportional hazard analysis assessing PFAS and OH-PCBs effects on the risk for confirmed disability worsening as well as the risk of transitioning from RRMS to SPMS.

### Regional concentration differences

Regional differences in PFAS and OH-PCBs concentrations were analyzed using the fully adjusted model described in the differential expression analysis section, including a term for the region. Sweden was divided into six larger regions. The division was based on how healthcare is organized in Sweden, with at least one university hospital in each region (Fig. 3A).

### PFAS and OH-PCB correlation with 25(OH)D3

To further analyze PFAS and OH-PCBs association with 25(OH)D3 levels, correlation analysis were performed. Spearman's rank correlation analysis was performed separately on log$_2$-transformed relative intensities for MS and HC subjects. Relative intensities below the detection limit of PFAS or OH-PCB compounds were imputed with half the lowest relative intensity. Correlations were adjusted for sex, sample collection month, and number of childbirths.

Further, Cox proportional hazard analysis was performed to assess the 25(OH)D3 effect on the risk for confirmed disability worsening. The analysis included the following covariates: age, sex, the time difference between diagnosis and sample time point, EDSS at the time of sample collection, current BMI, treatment, years of regular smoking, years of irregular smoking, years of indoor passive smoking, number of childbirths, and the month blood samples were collected. Moreover, Cox proportional hazard analysis was also performed to assess the 25(OH)D3 effect on the time to transition from RRMS to SPMS. The analysis included the following covariates: age, sex, the time difference between diagnosis and sample time point, current BMI, treatment, years of regular smoking, years of irregular smoking, years of indoor passive smoking, number of childbirths, and the month blood samples were collected. Finally, the two covariates, 25(OH)D3 levels and sample collection month, were also added to the Cox proportional hazard analysis assessing PFAS and OH-PCBs effect on risk for confirmed disability worsening as well as the time to transition from RRMS to SPMS.

### Reporting summary

Further information on research design is available in the Nature Portfolio Reporting Summary linked to this article.

## Data availability

All data generated and analysed during this study are included in this published article (and its Supplementary Information files/Source data). However, in accordance with Swedish laws on personal integrity and clinical data, as well as the Ethics Committee's decision, we are prohibited from making any personal data, including clinically-related variables, publicly accessible, even in an anonymized form. Data were, however, available upon ethical approval and application to the Swedish National Board of Health and Welfare (https://etikprovningsmyndigheten.se; the application procedure and instructions are provided in the link) and data access approval from the Swedish Neuro Registry (https://www.neuroreg.se/en/forskning/; the application procedure and conditions are provided in the link). The data may only be used for research and are not available for commercial use. Underlying data for all figures are provided in the Source Data file. Source data are provided with this paper.

## Code availability

All code related to the analyses in this study are available at https://github.com/caramba-uu/PFAS_and_OH-PCBs_in_MS.

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

## Acknowledgements

We thank all the participants of the EIMS cohort. We also acknowledge the Swedish Neuro Registries/Multiple sclerosis. This study was funded by the Swedish Research Council (2021–02814 (J.B.), 2021–02189 (K.K.), and 2024-03161 (K.K.)), the Swedish Society for Medical Research (J.B.), the Marianne and Marcus Wallenberg Foundation (J.B.), NEURO Sweden (K.K.), Region Uppsala (ALF-grant and R&D funds) (K.K. and J.B.), FORMAS (2020–01267 and 2023-00905) (K.K.), Åke Wiberg foundation (K.K. and H.C.), Forte (2024-01410) (I.E.), and The Swedish Foundation for MS Research (A.V. and I.E.). This work was supported by the SciLifeLab & Wallenberg Data Driven Life Science Program, Knut and Alice Wallenberg Foundation (grants: KAW 2020.0239 and KAW 2017.0003), and by the National Bioinformatics Infrastructure Sweden (NBIS) at SciLifeLab. The funding agencies had no influence on the design and conduct of the study; collection, management, analysis, and interpretation of the data; preparation, review, or approval of the manuscript; and decision to submit the manuscript for publication.

## Author contributions

The EIMS cohort was conceptualized by I.K., T.O., A.K.H., and L.A. The methods were developed by K.K., H.C., I.E., A.V., E.F., and P.E. A.A.G. and A.O.L. conducted the experimental analysis. A.V. implemented the methods and analysis. The results were interpreted by K.K., J.B., I.E., V.G., T.Å., and A.V. A.V., I.E., Y.N., H.C., O.S., A.A.G., A.O.L., J.B., and K.K. wrote the manuscript, and all the authors were involved in the review and editing process. All authors read and approved the final manuscript.

## Funding

## Competing interests

T.O. has received honoraria for lectures/advisory boards from Biogen, Novartis, Merck, and Sanofi and unrestricted MS research grants from the same companies, academic grants from The Swedish Research Council, the Swedish Brain Foundation, Knut and Alice Wallenberg Foundation, and Margaretha af Ugglas Foundation. L.A. has received honoraria for lectures Biogen, Merck, and Teva, as well as academic grants from The Swedish Research Council, the Swedish Brain Foundation, the Swedish Council for Health, Working Life and Health, Region Stockholm, and the insurance company AFA. IK has received honoraria for lectures from Merck as well as research collaborative support from Neurogene INC for an unrelated project. The remaining authors declare no competing interest.
