## [Peer Review File · Nature Communications]

Associations of PFAS and OH-PCBs with Risk of Multiple Sclerosis Onset and Disability Worsening

Corresponding Author: Dr Kim Kultima

Version 0:

Reviewer comments:

Reviewer #1

(Remarks to the Author)

This is a well written manuscript on issues that are of growing health importance. The paper would be enhanced by a focus on disease onset also.

- 1) Page 2, line 48, any more indices as proxies for immune activity? e.g. hay fever symptoms. No evidence immunomodulation is the mechanism, epidemiology prospective association only. What is the relationship to onset?
- 2) Introduction general, if lower levels at MS onset is that merely a marker of higher parity (a protective risk factor)?
- 3) Introduction general, need to understand onset as well. The relationship between PFAS levels and MS onset is also of interest.
- 4) Page 5, line 99, any evidence on PFAS? If not say so.
- 5) Page 5, line 112, discuss Ammitzbøll 2019 (PMID: 30852181) and Steenland 2013 (PMID: 23735465) (cross sectional)
- 6) Page 7, line 148, did controls have any follow up?
- 7) Page 7, line 156, "the seven targeted OH-PCBs" mention this in Abstract and provide results.
- 8) Page 10 Figure 1, how did female PFAS levels vary by parity (should be inverse). Check no such association for males.
- 9) Page 12 line 211, "meticulous matching" this sentence should be rewritten, for many questions matching by residence could be overmatching. For example if residence was an antecedent of the putative exposure.
- 10) Page 13, line 226, "and number of childbirths". Number of childbirths is a determinant of PFAS levels in women and inversely associated with MS. This should not be routinely adjusted for but considered more closely analytically. In any case, how was number of childbirths adjusted for in men?
- 11) Page 15, line 262 – 264, this is still not clear to me. Is the first EDSS at baseline? Did you use a three category variable?
- 12) Discussion general, before concluding PFAS is acting through immunomodulatory, do you have anything else that is a marker of immunomodulation e.g. active rhinitis (allergic?).
- 13) Discussion general, PFAS half life is not short but long. In females high parity decreases risk of MS and alters PFAS body stores. Please discuss and provide these results here.
- 14) Page 20, line 346, "immunotoxicity" not specific what does it do to the immune system and CNS including preclinical work.
- 15) Page 21, line 380, but you have the capacity to do so comparing cases and controls at baseline. Please do this.
- 16) Page 22, line 416, in discussion please provide more magnitudes of effect.
- 17) Page 22, line 419, "decreased PFOS concentrations in MS compared to HC subjects" provide Odd Ratio.
- 18) Page 22, line 426, How much post diagnosis? A baseline case control evaluation would still be important. This study has reported other things regarding onset e.g. smoking (PMID: 24146047), low sun exposure (PMID: 31844981) and insufficient sleep during adolescence (PMID: 36690431)
- 19) Page 22, line 442, "initially designed to evaluate environmental factors effect on MS, not the toxicological effects of environmental contaminants occurring before disease onset". In environmental epidemiology there is usually not a huge separation from toxic exposures such as smoking and ones obtained from biomarkers such as PFAS. In a rare disease such as MS incidence case control studies are one of the few ways to study such exposures where cohorts are not feasible.
- 20) Page 23, line 446, "another limitation is that no data is available regarding childbirth events or the number of menstruations after disease onset" but maternal parity prior to onset in another part you said you matched on childbirth number. Please explain.
- 21) Page 23, line 453, this is more an issue for absolute levels if lower in females. That is the paper should give more information on absolute differences between males and female particularly at onset and the contributions of parity/number of

children on this.

22) Page 24, line 481, "residential area" discuss how this may have been a problem regarding case-control comparison. What component of the variation of PFAS is linked to residential area? What is the effect estimate with residential criteria removed?

23) Page 26, line 532, "number of childbirths" so you do have this data here. Could you examine regarding PFAS levels.

24) Page 26, line 560, do you have relapse as an outcome?

25) Supplementary information, Supplementary Table S4 – Here are the onset results and other findings such as the PFHpA result should be put into main text. The 4-OH-CB187 and 3-OH-CB153 results are very important and require multivariable analysis, put in title. The article title should also refer to these pesticides. Population wide exposure to pesticides is increasing so this is a very important area to highlight and discuss.

26) Supplementary information, Supplementary Table S5, RRMS vs HC more adverse effects in females, p values for difference in effect? PMS vs HC 4-OH-CB187 and 3-OH-CB153 results go with literature regarding oxidative toxins in PPMS, please discuss.

27) Supplementary information, lines 77 and 78. Please remove tables and figures such as these if previously reported.

Reviewer #2

(Remarks to the Author)

Review of Per- and polyfluorinated 1 substances (PFAS) and risk for confirmed disability worsening in multiple sclerosis NCOMMS-24-24226

Summary: This is an interesting idea and a reasonably written study about PFAS internal exposure and its association with MS and especially with longitudinal MS clinical course. There are many design strengths such as case-control design nested in MS cohort. Three topics of confounding have been insufficiently considered or not considered. I do not know if consideration of any will necessarily have impact on the findings It is just that consideration of logical sources of confounding bias, or mention of a limitation that cannot be addressed, improves the work.

1. Potentially Insufficiently considered: It is unclear if the topic of confounding by body habitus has been sufficiently considered. Is BMI a good measure of relative body fat composition? There are several articles showing that Serum PFAS are inversely correlated with body fat.

Although not all studies show the same thing, the best performed studies in the most relevant age group suggest that some PFAS, notably the alkylates such as PFOA and PFNA are inversely associated with measures of body fat. (See for example Lind et al., PMID: 35074350 DOI: 10.1016/j.envres.2022.112677)

This is a possible confounder in terms of noncausal association (The data about body fat and PFAS are squishy). It can be addressed within the data for % body fat if available, a more specific measure than BMI or else mentioned as a possible study limitation that pertains to this work (and to many PFAS studies).

2. Not considered: In muscle wasting, serum albumin can be lower. MS is a well-known cause of muscle inactivity and wasting. This is not just a theoretical consideration. In meta-analysis, serum albumin is inversely correlated with MS PMID 32982663 PMID 32982663 PMID 32982663

In contrast, multiple studies show that serum PFAS are directly correlated with serum albumin. (This is so well-known that references are not provided).

Together, these data present a potential cause of reverse causation. The authors may wish to consider if serum albumin could provide alternative, non-causal explanations for associations. My thought is that the associations could easily be robust to such considerations, yet there is no knowing if the questions are not asked and confidence in the study and its design will be higher if they are asked. (A simpler alternative is to only mention that this type of noncausal reason for association was not addressed, and could be a source of reverse causation, a problem also mostly missed by at least one of the cited references for another disease state(39)).

3. Insufficiently considered: Matching on current residence at diagnosis makes sense for serum PFAS, but it is imperfect for MS if there are immigrant participants. Foreign immigrants will generally come from lower (nearer equator) latitudes and have lower a priori risk for MS than those born in Sweden. (This source of potential bias is not addressed by measures of vitamin D at the time of diagnosis.)

In addition, there are multiple demonstrations of different serum PFAS in specific ethnic populations, for example as seen in NHANES data across race/ethnicities and as seen in serum PFAS for residents of different countries. If there are immigrants in the study (not discernable from data I saw), they could be a priori more likely to be controls, and immigrants could have different serum PFAS than native populations (as shown in data from other nations). This might lead to a different type of bias, again depending on the actual number of immigrants in the study population. A sensitivity test could exclude foreign-born cases and controls.

Of these three potential sources of bias, the reviewer considers the albumin issue to be the most important. As MS worsens, and if the meta-analysis data are representative of the actual relationship of MS to serum albumin, reverse causation could

be an alternative explanation for key findings.

Abstract: The abstract is well written.

The authors could help the reader understand the point of the abstract better by clarifying syntax. It can be confusing to think about the inverse of worsening associated with a higher exposure (a double negative linked to a positive). From the abstract alone, the reviewer thinks but was not initially sure that the language means that higher serum PFAS is associated with better MS results over time (less disability). I needed to run to the tables to be sure my reading of the abstract was correct. Could the authors find some simple language solution so that the reader is not faced with solving the syntax problem of inverse to something bad (better or at least less awful outcome) associated with more internal exposure?

If PCBs are an important part of the study, and it is not so clear, then they should be part of the abstract.

Introduction: The introduction is mostly well written.

Line 99 Quoting: Moreover, low vitamin D levels have also been associated with an increased risk of 100 MS.7,8

The reviewer's impression is that this hypothesis remains controversial. It is fine to mention it, but some acknowledgement of uncertainty is suggested, unless the authors can make an argument for definitive data. Other risk factors currently have wider support, especially wider distribution of age of first encounters with common infectious diseases, which is also considered to be an alternative explanation (a more important explanation than vitamin D) for the childhood-latitude distribution of MS.

Line 118 Quoting: the current knowledge and findings of PFAS associations with autoimmune disease are scarce and, at large, 119 often have contradicting results in 120 humans¹⁴

This is explicitly a paper about a potentially autoimmune condition, so the reader may expect something more thorough. There is quite a bit of information about ulcerative colitis, for example, and increasing information about pregnancy-associated hypertension/preeclampsia (a specific setting). For these, and most evidence suggests an association of the condition to PFOA exposure and the concern that the PIH is all about reverse causation seems to be dispelled). For autoimmune thyroiditis, the data are complex and hard to interpret.

Line 122 Quoting Polychlorinated biphenyls (PCBs), another type of endocrine-disrupting chemicals, are, by contrast, well¹²³ established as carcinogens and are known to have other harmful effects on human health, including¹²⁴ developmental toxicity, neurotoxicity, and immunotoxicity. Consequently, many countries have restricted ¹²⁵ or banned their production and use since the 1970s, though they persist in the environment^{15,16}. In humans, ¹²⁶ hydroxylation of PCBs can occur, resulting in OH-PCBs, and the toxicity of PCBs has been suggested to originate partly from their hydroxylated metabolites. Moreover, OH-PCBs have a range of toxic traits not ¹²⁸ observed in parent PCBs, such as inhibition of mitochondrial respiration, disruption of thyroid hormone¹²⁹ activity and estrogenic activity, and generation of reactive oxygen species¹⁶.

The purpose of the paragraph about PCBs only is not that clear and the reasonable interpretation of these sentences and the thought about "in contrast" is that PFAS are not known carcinogens. IARC considers PFOA a class 1 and PFOS a class 2a carcinogen.

Line 131: The primary aim of this matched case-control study was to investigate the concentrations of PFAS and OH¹³² PCB compounds in individuals recently diagnosed with MS and their matched controls (HC). Additionally, ¹³³ we aimed to explore whether the concentrations of PFAS and OH-PCB compounds influenced the course. ¹³⁴ of the disease. We investigated the association between these chemicals and vitamin D₃ levels and explored ¹³⁵ if the presumed protective effects of vitamin D₃ are linked to PFAS and OH-PCB exposure.

The abstract and introduction should be aligned. The abstract does not hint that PCBs are a crucial part of the MS consideration. Are they?

Methods

The methods come after the discussion. Time-wasting for reviewers and readers. That may be a feature of this journal and not a bug of this paper, in which case the authors have merely complied with anachronistic requirements.

The use of BMI at two points in time is a valuable approach. However, noted above, relative body fat could change rapidly in early disease and BMI need not reflect that well in a disease state characterized by muscle-wasting. It appears possible to likely that relative body fat is inversely related to serum PFAS. IF there is no way to address this issue (it is rarely addressable with available data), it can be mentioned as a study limitation.

The potential impacts of immigration on serum PFAS are partially addressed with the mononucleosis question, but it is the

greater likelihood of absence of such diagnosis that is relevant for immigrants, who may be overrepresented in controls to the degree they are in the study. This deserves a sensitivity test if possible.

MS and albumin (inversely associated with MS in meta-analysis and directly associated with serum PFAS for obvious reasons of protein binding) is a reverse causation topic mentioned above. Because the logical reason for this finding is disease state, this may also be pertinent to disease course and not just to initial diagnosis.

Results:

The results are reasonably presented, interesting, and plausible. Overlooked implications that could alter their presentation follow under discussion.

The importance and relevance of the short-chain short half-life serum PFAS values is not that clear. The range is narrow, and the comparisons appear forced to this reviewer. If they are retained as an important part of the results and discussion, the limitations of relatively narrow ranges (and of course the frequent nondetects) should be more clearly discussed. This topic appears in the methods, and then it feels like it is minimized in results and especially discussion.

Discussion:

352-356, 428-430 Precise age-year specific serum PFAS are relevant to the diagnosis of MS, as the authors note, and are available in an NHANES study from age 12 onwards, which might assist readers to understand the authors' point.

In addition, since the sexes reconverge (at least for the four most common serum PFAS) at around age 52 it is possible that sex considerations do not matter for diagnoses at later ages. (Up to the authors if this is helpful).

435-437 457-59 Meticulous residential matching is indeed a strength for serum PFAS comparisons, but it does little for the possible effect of immigration on MS diagnosis and any differences in either serum PFAS or MS course in immigrants vs native-born. The authors are likely to be aware that Immigration is a very important topic for MS specifically, and it is not addressed in the paper. Mentioned above, a sensitivity test limiting comparisons to natives, or a demonstration that the contribution of immigrant participants is a priori de minimis, or a mention of the limitation is commended to the authors to improve the paper.

The authors can consider whether the % body fat predicts both increased MS severity and also lower serum PFAS is as plausible as the reviewer suspects, and not fully adjusted by BMI.

369-374 and also 415-16, 446-48 Please see previous discussion about albumin, and also consider hemoglobin (also a protein that binds to PFAS and predicts serum PFAS!!) The authors are generally aware of this topic as regards menstruation, but it is not accounted for in the study design and adjustments. To what degree are the findings reverse causation, including the disease severity findings? One of the cited articles (39) is from a group which consistently ignored this topic. Clinically, albuminuria is common in diabetics and hypertensives, and often one of the overlooked presenting findings in diabetes. Lower serum albumin can be present even in the absence of albuminuria. (The group responsible for reference 39 wrote about the protective effect of PFAS on diabetes, a conclusion now regarded as a cautionary tale of overlooked reverse causation. It was subsequently shown that this is due in whole or part to proteinuria and lower serum protein that follows from diabetes.) To the degree possible, it is worth considering if this same kind of reverse causation might affect MS which has a meta-analysis showing inverse association to serum albumin. (This is likely to be less strong than in diabetes, but it is the same topic, and it will be more convincing to readers aware of the topic if it is considered.)

376-378 The association of PFAS to CVD is far more complex than presented and has several recent well-done longitudinal studies which provide a more nuanced view of this topic than the authors' have presented. This topic can either be omitted or expanded to reflect the emerging longitudinal findings more accurately.

391-96 The authors present vitamin D associations as definitively causal (they present contravening clinical trial data, and the arguments against causation include that and more). The language should be altered to better reflect the controversy, or else the authors can better defend their causal language while accounting for more of the opposing perspective.

415-416 The reviewer perspective is that this would be a reasonable place to discuss how and why the disease severity findings could relate, in whole or in part, to reverse causation based on protein binding to albumin and hemoglobin. In contrast, the same considerations might strengthen the possibility of initial causation, about which the authors are appropriately understated (and could mention more).

Reviewer #3

(Remarks to the Author)

I read this paper with interest. The authors should be commended on the thorough attempt to tackle this complex question using a unique cohort. As they state, the contribution of PFAS to autoimmune disease (and immunity in general) is generally poorly understood, but has the potential to deepen our understanding around both these compounds and also disease pathogenesis.

I did however have some areas where I felt that further clarification was required.

That the concentrations of these substances are typically not independent of each other is an important point, and needs to be made more clearly in the results. I note that the authors state that this violates the assumptions for multiple testing, yet it is not clear from the methods how the inherent concerns around multiple testing of associated exposures has been dealt with. I thought that this needed more clarity, as with a large range of compounds tested and few significant results (with significance in both directions of effect), this becomes a crucial consideration. Following on from this, the finding that the direction of effect in results reaching statistical significance is not consistent between males and females, and between compounds becomes of some concern, and I did not feel this was developed in depth in the discussion. There also appear to be differences when considering susceptibility (MS vs HC) and severity (CDS), however I found the discussion on why this might be slightly limited.

In the methods, I note that DMT was not included in a mediator of disability outcomes. I would be interested as to why this decision was taken when other potential mediators of confirmed disability worsening were included. Given that DMT likely has a major influence on outcomes, and that DMT access is likely subject to substantial variation, for me this would be an important exposure to consider.

Additionally, I note that in the methods it is stated that those patients without EDSS within 3m of sample onset were excluded from the Cox analysis looking at CDW. In total only 551 patients were included in the CDW analysis – were these participants demographically identical to the whole cohort? There is no data presented that I could see on the phenotypes of those included in this analysis.

Follow up time in the CDW cohort is also unclear. I note that it is stated that FU was up to 18 years. What was the mean, median range? How was differing FU time dealt with in the Cox model, as this will depend on the variation in FU times.

In the results and discussion, I would disagree with the assertion that these data show an effect of 25(OH)D3 on CDW. They show an association, which may result from reverse causation (those with worse disability have less sunlight exposure due to disability/behavioural change). The language around association vs causation is appropriate through most of the paper, but this did stand out.

Results

Table 1

I was unclear what % means for EDSS – is it the % for which one measure was available? Or the % with an EDSS available within a given time range of biosampling? I think that it was % with EDSS within 3m of biosampling, but this was only clear in the methods and not apparent when reading the table. One would normally give median rather than mean EDSS as this is a non-linear non-continuous measure.

I note that EDSS is used rather than correcting for age with an ARMSS. Given the variation in both PFAS and EDSS with age is EDSS the right measure to use, or would applying an age correction be more rigorous?

Only 13.7% of RRMS on treatment seems low – again is this at baseline? As there appears to be FU as part of this study this needs clarification and how graded exposure (in terms of both time and DMT efficacy) is dealt with.

For irregular smoking, is this just of those who do not declare regular smoking?

The measure used to define alcohol consumption is not clear to me in the table, although I note it is given in methods.

Figure 3c

I found it difficult to appreciate the differences between regions given the scale on the graph. The low (highly significant) p values were notable given the small differences illustrated. I note that multiple corrections were made but interestingly not disease status, although it appears that MS and HC were pooled? Could this have driven some of the findings?

Figure 5

Is 3-OH-Cb153 associated with positive association with disability worsening? It appears to be bordering on significance from the plot. This was not really mentioned in results that I could see.

Version 1:

Reviewer comments:

Reviewer #1

(Remarks to the Author)

The revision is good. Could the title be clarified to be

From Disease Onset Risk to Disability Worsening not 'Disease Risk' which is too non-specific. Otherwise looks good.

Reviewer #2

(Remarks to the Author)

The authors have been attentive to reviewers comments. It is a pleasure to see this work progress.

Reviewer #3

(Remarks to the Author)

This paper has improved substantially, particularly in terms of clarity of how some of the results are presented. I also agree with the other reviewers regarding the value of the additional analyses. I have no further comments or suggestions.

Point-by-point response to reviewers

We acknowledge the constructive comments made by the three reviewers and have done our best to address them. While this has made the study more extensive, we believe this has also strengthened its value and content. We have also added an author, Anders Olof Larsson, who has greatly contributed to one of the major points raised by Reviewer #2.

In brief,

- As one of the major points by Reviewer #1, we have performed made more in-depth analyses regarding the role of childbirth. We have now compared PFAS and OH-PCBs concentrations among females who have given birth versus females who have not. We found significantly lower concentrations of eight PFAS and two OH-PCBs in females with biological children. A similar analysis was also performed in males, demonstrating no significant differences in PFAS and OH-PCBs concentrations.
- As a major point by Reviewer #2, we have now measured serum albumin in all the 1,814 individuals in the study. The serum albumin concentrations have been analyzed with respect to the risk of confirmed disability worsening (CDW) to investigate the possibility of reverse causation in our results. We found no associations between albumin concentrations and risk for CDW. We have also performed sensitivity analyses, including albumin, in the models to assess risks of PFAS/OH-PCBs and CDW, and we found that the previously observed inverse associations remained. Moreover, 3-OH-CB153 demonstrated a significant positive association with CDW in males.
- As requested by Reviewers #1 and #2, we have performed additional analysis regarding PFAS and OH-PCBs' association with the risk of developing multiple sclerosis (MS), which has also been added to the study. We found two OH-PCBs (4-OH-CB187 and 3-OH-CB153) to be significantly associated with an increased risk of developing MS. Furthermore, sensitivity analysis was performed by taking immigration status (Swedish or Nordic-born status) into account regarding the risk of developing MS and the pairwise comparison between MS types and controls, demonstrating similar results.
- As requested by Reviewers #1 and #2, we have also extended the introduction and discussions with more information on previous studies on PFAS and OH-PCBs in the context of autoimmune diseases and *in vitro* and *in vivo* studies. As pointed out by the Reviewers, we have also rephrased the language with respect to associations and possible reverse causation.

- As requested by Reviewer #3 we have also performed a sensitivity analysis, in which treatment during the follow-up time was included in the Cox proportional hazard analysis.
- Finally, we have also changed the title of the manuscript, as requested by Reviewer #1, to include OH-PCBs in the title.

Reviewer #1:

This is a well written manuscript on issues that are of growing health importance. The paper would be enhanced by a focus on disease onset also.

- 1. Page 2, line 48, any more indices as proxies for immune activity? e.g. hay fever symptoms. No evidence immunomodulation is the mechanism, epidemiology prospective association only. What is the relationship to onset?**

Reply: In this study, we do not have access to data regarding fever etc. that could be a proxy for immune activity. As this is an epidemiological study, the mechanism behind PFAS and OH-PCBs immunomodulatory properties has not been studied. We have now rephrased the sentence regarding PFAS possible immunosuppressive effects (page 3, lines 48-51) as well as added the results regarding onset (page 3, lines 40-42, and page 17, lines 295-309).

“These results show previously unknown associations between OH-PCBs and the risk of developing MS, as well as the inverse associations between PFAS exposure and the risk of disability worsening in MS.”

“Moreover, two OH-PCBs (4-OH-CB187 and 3-OH-CB153) were associated with a significant ($P < 0.05$) increased risk of developing MS, regardless of sex and immigration status.”

“To explore potential associations between exposure to individual PFAS compounds or OH-PCBs, and the risk of developing MS, logistic regression was used. Short-chain PFAS have been excluded from the risk of developing MS analysis due to their short half-lives. Two models were used, a base model adjusting for age, sex, BMI at sample collection, and BMI at the age of 20, and a fully adjusted model where smoking habits, sun exposure, history of infectious mononucleosis, number of childbirths, and Swedish-born or not were included. This study found 4-OH-CB187 and 3-OH-CB153 to be significantly ($P < 0.05$) associated with increased odds of MS in the two models (Table 2, Supplementary Table S19) regardless of sex. A doubling of serum 4-OH-CB187 concentrations increased the

odds of MS by 8.1% (OR = 1.081, 95% CI: 1.019-1.148, P = 0.010), while 3-OH-CB153 increased the odds by 5.5% (OR = 1.055, 95% CI: 1.011 - 1.101, P = 0.015). Similar results were also observed when dividing by sex. The model estimates, and ANOVA P values for the models are presented in Supplementary Tables S20 to S25. A sensitivity analysis was performed, demonstrating similar results when adjusting for Nordic heritage instead of Swedish (Supplementary Tables S26 to S29). Further, sensitivity analysis was also performed in relation to the residential area criteria, removing this criteria did not impact the results (Supplementary Tables S30 to S33.)“

2. Introduction general, if lower levels at MS onset is that merely a marker of higher parity (a protective risk factor)?

Reply: The number of completed births, which is the data we have access to, has been taken into account in the model. Also, there was no significant difference in the number of births between HC and MS when comparing females who have given birth. Among HC there is a slightly higher percentage of females who have given birth compared to MS. In spite of taking this into account, there are significant differences between MS and controls. However, we can, of course, not be sure that this completely covers this potential protection. This has now also been added as sentence to the limitations of the study (page 30, lines 603-608).

“Regardless, adjusting for the number of childbirths in the model may not fully adjust for the female-specific PFAS and OH-PCBs excretion routes. We can thus not exclude the possibility that the difference in childbirth between HC (71.7%) and MS (62.4%) may, to some extent, contribute to observed differences. However, among those who have given birth, there is no significant difference in the number of childbirths between HC (mean 2.1) and MS (mean 2.0).”

3. Introduction general, need to understand onset as well. The relationship between PFAS levels and MS onset is also of interest.

Reply: We agree that the relationship between PFAS and OH-PCB levels and MS onset is of great interest as well. Analysis regarding this has been added to the study (page 17, lines 295-309). In brief, 4-OH-CB187 and 3-OH-CB153, were both associated with an increased risk of developing MS. No such risks were found for PFAS related compounds.

“To explore potential associations between exposure to individual PFAS compounds or OH-PCBs, and the risk of developing MS, logistic regression was used. Short-chain PFAS have been excluded from the risk of developing MS analysis due to their short half-lives. Two models were used, a base model adjusting for age, sex, BMI at sample collection, and BMI at the age of 20, and a fully adjusted model where smoking habits, sun exposure, history of infectious mononucleosis, number of childbirths, and Swedish-born or not were included. This study found 4-OH-CB187 and 3-OH-CB153 to be significantly ($P < 0.05$) associated with increased odds of MS in the two models (Table 2, Supplementary Table S19) regardless of sex. A doubling of serum 4-OH-CB187 concentrations increased the odds of MS by 8.1% ($OR = 1.081$, 95% CI: 1.019-1.148, $P = 0.010$), while 3-OH-CB153 increased the odds by 5.5% ($OR = 1.055$, 95% CI: 1.011 - 1.101, $P = 0.015$). Similar results were also observed when dividing by sex. The model estimates, and ANOVA P values for the models are presented in Supplementary Tables S20 to S25. A sensitivity analysis was performed, demonstrating similar results when adjusting for Nordic heritage instead of Swedish (Supplementary Tables S26 to S29). Further, sensitivity analysis was also performed in relation to the residential area criteria, removing this criteria did not impact the results (Supplementary Tables S30 to S33.)“

Regarding the introduction, we added some discussion on previous research investigating the relationship of PFAS with MS disease onset (page 7, lines, 124-135).

“PFOA exposure has been associated with an increased risk of developing ulcerative colitis²⁷. In addition, a recent study demonstrated an association between higher PFAS concentrations and a decreased risk of developing rheumatoid arthritis²⁸. To this date, no risk analysis has been performed concerning PFAS exposure and the risk of developing

MS. A cross-sectional study examining the relationship between autoimmune disease and estimated past exposure levels of perfluorooctanoic acid (PFOA), a common PFAS, demonstrated no significant association between PFOA levels and the incidence of MS²⁷. However, the PFOA concentrations were estimated from intake of contaminated drinking water, assuming low background exposure, and not measured. Additionally, a case-control study showed lower concentrations of PFOA in individuals with MS compared with controls, suggesting that PFOA exposure was not an important risk factor for MS, but no risk analysis was performed²⁹. To date, there are no studies investigating whether concentrations of PFAS have any role in disease progression in MS.”

4. Page 5, line 99, any evidence on PFAS? If not say so.

Reply: We discuss PFAS and the studies made on PFAS and MS in the following paragraph (page 7, lines, 124-135) and state that there is limited knowledge regarding PFAS role in the risk of developing MS.

“PFOA exposure has been associated with an increased risk of developing ulcerative colitis²⁷. In addition, a recent study demonstrated an association between higher PFAS concentrations and a decreased risk of developing rheumatoid arthritis²⁸. To this date, no risk analysis has been performed concerning PFAS exposure and the risk of developing MS. A cross-sectional study examining the relationship between autoimmune disease and estimated past exposure levels of perfluorooctanoic acid (PFOA), a common PFAS, demonstrated no significant association between PFOA levels and the incidence of MS²⁷. However, the PFOA concentrations were estimated from intake of contaminated drinking water, assuming low background exposure, and not measured. Additionally, a case-control study showed lower concentrations of PFOA in individuals with MS compared with controls, suggesting that PFOA exposure was not an important risk factor for MS, but no risk analysis was performed²⁹. To date, there are no studies investigating whether concentrations of PFAS have any role in disease progression in MS.”

5. Page 5, line 112, discuss Ammitzbøll 2019 (PMID: 30852181) and Steenland 2013 (PMID: 23735465) (cross sectional)

Reply: We agree that the introduction should include the mentioned articles. To address this issue we have added a discussion regarding the results presented in Ammitzbøll 2019 (PMID: 30852181) and Steenland 2013 (PMID: 23735465) (page 7, lines 127-135).

“A cross-sectional study examining the relationship between autoimmune disease and estimated past exposure levels of perfluorooctanoic acid (PFOA), a common PFAS, demonstrated no significant association between PFOA levels and the incidence of MS²⁷. However, the PFOA concentrations were estimated from intake of contaminated drinking water, assuming low background exposure, and not measured. Additionally, a case-control study showed lower concentrations of PFOA in individuals with MS compared with controls, suggesting that PFOA exposure was not an important risk factor for MS, but no risk analysis was performed²⁹. To date, there are no studies investigating if concentrations of PFAS have any role in disease progression in MS.”

6. Page 7, line 148, did controls have any follow up?

Reply: The controls did not have any follow-up. The MS cases were followed through the Swedish MS registry.

7. Page 7, line 156, “the seven targeted OH-PCBs” mention this in Abstract and provide results.

Reply: In the analysis, we targeted seven OH-PCBs, however only three OH-PCBs were detected in the study samples, and thus results can only be provided for those three compounds. We agree that the most important results related to these compounds should be included in the abstract and have addressed the issue accordingly (page 3, lines 40-42).

“Moreover, two OH-PCBs (4-OH-CB187 and 3-OH-CB153) were associated with a significant ($P < 0.05$) increased risk of developing MS, regardless of sex and immigration status.”

8. Page 10 Figure 1, how did female PFAS levels vary by parity (should be inverse). Check no such association for males.

Reply: In our understanding, parity refers to the number of times a female has been pregnant to gestational week 20 or longer. In this study, we do not have data on parity, however, we do have data on the number of children the women have given birth to prior to blood sample collection, referred to as the number of childbirths in the manuscript. All three analyses (risk of developing MS, MS vs HC, and risk for confirmed disability worsening) have been adjusted for the number of child births. To address this issue we have looked into the variation in PFAS levels between females with biological children and those without (see figure below). We included the same co-variables as previously (age, disease phenotype, BMI at the age of 20, BMI at sample time point, year of sample collection, years of regular smoking, years of irregular smoking, years of indoor passive smoking, and type of treatment). As expected females who had given birth to biological children had significantly ($P < 0.05$) lower levels of PFPeS, PFHpA, PFHxS, PFHpS, PFOA, PFOS, PFNA and PFDoA. The same analysis was also performed for males (see figure below), and no significant differences were found. These results have now also been added in Supplementary Table S2.

Figure 1. PFAS concentration levels between females with biological children and females without biological children. Controls and MS have been pooled.

Figure 2. PFAS concentration levels between males with biological children and males without biological children. Controls and MS have been pooled.

- 9. Page 12 line 211, “meticulous matching” this sentence should be rewritten, for many questions matching by residence could be overmatching. For example if residence was an antecedent of the putative exposure.**

Reply: We have now rephrased the part of the sentence with “meticulous matching” (page 14, lines 239-241).

“These findings show significant regional differences in PFAS and OH-PCB concentrations across Sweden, highlighting the need to consider regional differences when conducting a case-control study involving these compounds.”

- 10. Page 13, line 226, “and number of childbirths”. Number of childbirths is a determinant of PFAS levels in women and inversely associated with MS. This should not be routinely adjusted for but considered more closely analytically. In any case, how was number of childbirths adjusted for in men?**

Reply: Since childbirths is considered as one of the sex-specific exit routes for PFAS, the number of childbirths was set to 0 in males. See also comment 8 (Page 2, Figure 1) above, where we investigated more closely the variations in PFAS levels with childbirth/biological children in men and women.

- 11. Page 15, line 262 – 264, this is still not clear to me. Is the first EDSS at baseline? Did you use a three category variable?**

Reply: We have addressed this issue by providing a more detailed explanation of the methods (page 19, lines 320-322). The first EDSS is the closest measurement to the time point when the sample was taken (within 3 months), referred to as the baseline. We did not use a three-category variable. For confirmed disability worsening, we used the definition proposed by Kalincik et al. where the minimum EDSS increase required for confirmed disability worsening is defined in three different ways, depending on the individual's baseline EDSS. If the baseline EDSS is between 0.0 and 1.0, an increase of at least 1.5 points is needed to define confirmed disability worsening. If the baseline EDSS is between 1.5 and 5.0, a minimum increase of 1.0 points is required. For a baseline EDSS of 5.5 or

above, an increase of 0.5 points is necessary. Please see (PMID: 26359291, PMID: 37782515) for more information.

“Here, we defined confirmed disability worsening based on the baseline EDSS as the following: from EDSS 0.0 to 1.0 an increase with ≥ 1.5 points is required; EDSS 1.5 to 5.0 requires ≥ 1.0 points increase; and EDSS ≥ 5.5 requires ≥ 0.5 points increase^{35,36}.“

12. Discussion general, before concluding PFAS is acting through immunomodulatory, do you have anything else that is a marker of immunomodulation e.g. active rhinitis (allergic?).

Reply: We do not have any markers of immunomodulation. However, one of the possible explanations for PFAS inverse association with confirmed disability worsening is its immunosuppressive effects. We have now changed the language regarding this topic throughout the discussion.

13. Discussion general, PFAS half life is not short but long. In females high parity decreases risk of MS and alters PFAS body stores. Please discuss and provide these results here.

Reply: We agree that for most PFAS the half-life is long (years), whereas the shorter PFBS, PFPeS, and PFHpA typically have a half-life of a few of months, up to a year. We have now added the results on the relationship between PFAS and MS disease onset (page 17, lines 295-309). Regarding the issues with parity, PFAS levels, and MS disease onset, we have adjusted for this in the models as discussed above (point 8). We found no significant associations between PFAS and the risk of developing MS. However, we did find that two OH-PCBs were significantly associated with the risk of developing MS, this finding was consistent for males and females.

“To explore potential associations between exposure to individual PFAS compounds or OH-PCBs, and the risk of developing MS, logistic regression was used. Short-chain PFAS have been excluded from the risk of developing MS analysis due to their short half-lives. Two models were used, a base model adjusting for age, sex, BMI at sample collection, and BMI at the age of 20, and a fully adjusted model where smoking habits, sun exposure,

history of infectious mononucleosis, number of childbirths, and Swedish-born or not were included. This study found 4-OH-CB187 and 3-OH-CB153 to be significantly ($P < 0.05$) associated with increased odds of MS in the two models (Table 2, Supplementary Table S19) regardless of sex. A doubling of serum 4-OH-CB187 concentrations increased the odds of MS by 8.1% (OR = 1.081, 95% CI: 1.019-1.148, $P = 0.010$), while 3-OH-CB153 increased the odds by 5.5% (OR = 1.055, 95% CI: 1.011 - 1.101, $P = 0.015$). Similar results were also observed when dividing by sex. The model estimates, and ANOVA P values for the models are presented in Supplementary Tables S20 to S25. A sensitivity analysis was performed, demonstrating similar results when adjusting for Nordic heritage instead of Swedish (Supplementary Tables S26 to S29). Further, sensitivity analysis was also performed in relation to the residential area criteria, removing this criteria did not impact the results (Supplementary Tables S30 to S33).“

14. Page 20, line 346, “immunotoxicity” not specific what does it do to the immune system and CNS including preclinical work.

Reply: We have now changed “immunotoxicity” to “immunosuppressive effects” to clarify PFAS effects on the immune system. We have also expanded the discussion on PFAS effects on the immune system, including pre-clinical work, in the following paragraph (page 26, lines 465-469).

“Moreover, in rodents, a dose-responsive suppression of the T cell-dependent antibody response, an assay focusing on the humoral arm of the adaptive immune system, has been associated with PFOA and PFOS exposure^{45,46}. In vitro studies provide further evidence for PFAS immunomodulating effects through modulation of oxidative stress, Ca^{2+} -signaling, nuclear receptor, and cytokine levels (reviewed 2023)¹⁴.”

15. Page 21, line 380, but you have the capacity to do so comparing cases and controls at baseline. Please do this.

Reply: We have now added a logistic regression analysis investigating if PFAS or OH-PCBs are associated with the risk of developing MS (page 17, lines 295-309). In brief, 4-OH-CB187 and 3-OH-CB153 were both associated with an increased risk of developing MS. No such risks were found for PFAS-related compounds.

“To explore potential associations between exposure to individual PFAS compounds or OH-PCBs, and the risk of developing MS, logistic regression was used. Short-chain PFAS have been excluded from the risk of developing MS analysis due to their short half-lives. Two models were used, a base model adjusting for age, sex, BMI at sample collection, and BMI at the age of 20, and a fully adjusted model where smoking habits, sun exposure, history of infectious mononucleosis, number of childbirths, and Swedish-born or not were included. This study found 4-OH-CB187 and 3-OH-CB153 to be significantly ($P < 0.05$) associated with increased odds of MS in the two models (Table 2, Supplementary Table S19) regardless of sex. A doubling of serum 4-OH-CB187 concentrations increased the odds of MS by 8.1% ($OR = 1.081$, 95% CI: 1.019-1.148, $P = 0.010$), while 3-OH-CB153 increased the odds by 5.5% ($OR = 1.055$, 95% CI: 1.011 - 1.101, $P = 0.015$). Similar results were also observed when dividing by sex. The model estimates, and ANOVA P values for the models are presented in Supplementary Tables S20 to S25. A sensitivity analysis was performed, demonstrating similar results when adjusting for Nordic heritage instead of Swedish (Supplementary Tables S26 to S29). Further, sensitivity analysis was also performed in relation to the residential area criteria, removing this criteria did not impact the results (Supplementary Tables S30 to S33).“

16. Page 22, line 416, in discussion please provide more magnitudes of effect.

Reply: For this particular statement, we do not have a magnitude of effect. However, we could see a clear sex difference in PFAS association with confirmed disability worsening between men and women. To clarify, we have now changed the language in this sentence (page 29, line 557-560).

“The male sex has also been associated with a shorter time to disability landmarks regarding EDSS and progression^{67,68}, which could potentially explain why PFAS association to disease progression was mainly observed in male MS subjects.”

17. Page 22, line 419, “decreased PFOS concentrations in MS compared to HC subjects” provide Odd Ratio.

Reply: The cited article (Ammitzbøll 2019, PMID: 30852181) has not performed any risk analysis and we can therefore not provide any odds ratios. The article only states that the levels of PFOS were 17% lower in MS subjects compared to controls.

18. Page 22, line 426, How much post diagnosis? A baseline case control evaluation would still be important. This study has reported other things regarding onset e.g, smoking (PMID: 24146047), low sun exposure (PMID: 31844981) and insufficient sleep during adolescence (PMID: 36690431)

Reply: The refereed articles are based on self-reported lifestyle habits from a survey looking at lifestyle habits (up to 10-20) years prior to diagnosis. The current study is based on blood samples taken post-diagnosis. Blood samples from individuals with RRMS were taken 8.9 ± 17.4 months after diagnosis in females and 8.8 ± 15.3 in males. While for individuals with a progressive form of MS blood samples were collected 13.6 ± 39.7 months after diagnosis in females and 10.5 ± 14.2 months in males. Due to PFAS and OH-PCBs long half-lives we are still able to perform analysis regarding their association with risk of developing MS. We have now included an analysis regarding the risk of developing MS (page 17, lines 295-309), as proposed. Please see point 15.

In accordance with the aforementioned studies, we also see, based on the covariates in the logistic regression (Supplementary Tables S23 to S25) that e.g. smoking significantly increases the risk of MS (as demonstrated in PMID: 24146047) and high sun exposure decreases the risk of MS (as demonstrated in PMID: 31844981).

“To explore potential associations between exposure to individual PFAS compounds or OH-PCBs, and the risk of developing MS, logistic regression was used. Short-chain PFAS have been excluded from the risk of developing MS analysis due to their short half-lives. Two models were used, a base model adjusting for age, sex, BMI at sample collection, and BMI at the age of 20, and a fully adjusted model where smoking habits, sun exposure, history of infectious mononucleosis, number of childbirths, and Swedish-born or not were included. This study found 4-OH-CB187 and 3-OH-CB153 to be significantly ($P < 0.05$) associated with increased odds of MS in the two models (Table 2, Supplementary Table S19) regardless of sex. A doubling of serum 4-OH-CB187 concentrations increased the odds of MS by 8.1% ($OR = 1.081$, 95% CI: 1.019-1.148, $P = 0.010$), while 3-OH-CB153 increased the odds by 5.5% ($OR = 1.055$, 95% CI: 1.011 - 1.101, $P = 0.015$). Similar results were also observed when dividing by sex. The model estimates, and ANOVA P values for the models are presented in Supplementary Tables S20 to S25. A sensitivity analysis was performed, demonstrating similar results when adjusting for Nordic heritage instead of Swedish (Supplementary Tables S26 to S29). Further, sensitivity analysis was also performed in relation to the residential area criteria, removing this criteria did not impact the results (Supplementary Tables S30 to S33).“

19. Page 22, line 442, “initially designed to evaluate environmental factors effect on MS, not the toxicological effects of environmental contaminants occurring before disease onset”. In environmental epidemiology there is usually not a huge separation from toxic exposures such as smoking and ones obtained from biomarkers such as PFAS. In a rare disease such as MS incidence case control studies are one of the few ways to study such exposures where cohorts are not feasible.

Reply: We agree and have now rephrased the sentence (page 30, lines 586-587). We have also now added an analysis regarding the risk for developing MS to the study (Results, section *Exposure to OH-PCBs are associated with an increased risk of developing MS*).

“The data used in this study (Epidemiological Investigation of Multiple Sclerosis (EIMS) cohort) was initially designed to evaluate environmental factors' effect on MS.”

20. Page 23, line 446, “another limitation is that no data is available regarding childbirth events or the number of menstruations after disease onset” but maternal parity prior to onset in another part you said you matched on childbirth number. Please explain.

Reply: We apologize for our mistake. Yes, we do have data on number of childbirths prior to sample collection. For clarification we did not match on childbirth numbers, the analysis has been adjusted for this factor. A limitation of this study is the lack of data regarding menstruations, as it serves as an important PFAS exit route in females.

21. Page 23, line 453, this is more an issue for absolute levels if lower in females. That is the paper should give more information on absolute differences between males and female particularly at onset and the contributions of parity/number of children on this.

Reply: We agree and have now rephrased the sentence as well as added more information regarding PFAS concentration differences between males and females and depending on childbirths. For a more detailed answer, please see point 8.

As expected females who had given birth to biological children had significantly ($P < 0.05$) lower levels of PFPeS, PFHpA, PFHxS, PFHpS, PFOA, PFOS, PFNA, and PFDoA. The same analysis was also performed for males (see figure below), and no significant differences were found. These results have now also been added in Supplementary Table S2.

22. Page 24, line 481, “residential area” discuss how this may have been a problem regarding case-control comparison. What component of the variation of PFAS is linked to residential area? What is the effect estimate with residential criteria removed?

Reply: When removing the residential area criteria similar trends can still be observed, however they typically become non-significant in females while the same effect is seen in males (Supplementary Tables S15-S18). This is an effect of the large variation in PFAS levels observed throughout Sweden.

23. Page 26, line 532, “number of childbirths” so you do have this data here. Could you examine regarding PFAS levels.

Reply: We acknowledge that the number of childbirth data that we have has not been described clearly enough. We have data on the number of childbirths prior to blood sample collection. To address this issue raised by the reviewer, we have examined how PFAS levels vary depending on whether a person has biological children or not (see results point 8).

24. Page 26, line 560, do you have relapse as an outcome?

Reply: We did not use relapse as an outcome since confirmed disability worsening is considered a more robust measurement of disease worsening.

25. Supplementary information, Supplementary Table S4 – Here are the onset results and other findings such as the PFHpA result should be put into main text. The 4-OH-CB187 and 3-OH-CB153 results are very important and require multivariable analysis, put in title. The article title should also refer to these pesticides. Population wide exposure to pesticides is increasing so this is a very important area to highlight and discuss.

Reply: The results referred to are for the mixed-linear regression analysis demonstrating concentration differences in individuals with MS and HC subjects, these results are also presented in *Figure 4* in the main article text. The models have been adjusted for the following covariates: age, sex, BMI at the age of 20, BMI at sample time point, year of sample collection, years of regular smoking, years of irregular smoking, years of passive smoking, number of childbirths, and type of treatment (Supplementary Tables S3-S6).

Regarding the associations to disease onset, we have now also added logistic regression analysis examining PFAS and OH-PCBs association with risk of developing MS (Results, section *OH-PCB exposure is associated with an increased MS risk*). The logistic regression model has been adjusted for: age, sex, BMI at sample time point, BMI at the age of 20, number of childbirths, regular smoking, irregular smoking, history of infectious mononucleosis, sun habits, born in Sweden or not, and residential area treated as a random effect (Supplementary Tables S23-S25).

We have now changed the title to include OH-PCB and updated the abstract with respect to develop MS in association with 4-OH-CB187 and 3-OH-CB153.

26. Supplementary information, Supplementary Table S5, RRMS vs HC more adverse effects in females, p values for difference in effect? PMS vs HC 4-OH-CB187 and 3-OH-CB153 results go with literature regarding oxidative toxins in PPMS, please discuss.

Reply: Previous Supplementary Table 5, now Supplementary Table S6, shows concentration differences between MS and HC at baseline for males and females separately. However, from these results, it is not possible to draw any conclusions regarding the adverse effects of PFAS or OH-PCBs, as they only describe concentration differences. We have not done any significant testing regarding the differences in effects.

Regarding the oxidative toxins in PPMS, we have not been able to find the exact literature referred to. We agree that OH-PCBs and PCBs can cause oxidative stress, and that is one of the mechanisms behind their neurotoxicity, which are important features in disease progression in PMS (PMID: 30976975 and PMID: 27829982).

27. Supplementary information, lines 77 and 78. Please remove tables and figures such as these if previously reported.

Reply: These results for this cohort have not been previously reported. However, results such as these that can be considered as well-known and have been previously reported by other groups have been removed from the supplementary information.

Reviewer #2:

Summary: This is an interesting idea and a reasonably written study about PFAS internal exposure and its association with MS and especially with longitudinal MS clinical course. There are many design strengths such as case-control design nested in MS cohort. Three topics of confounding have been insufficiently considered or not considered. I do not know if consideration of any will necessarily have impact on the findings It is just that consideration of logical sources of confounding bias, or mention of a limitation that cannot be addressed, improves the work.

- 1. Potentially Insufficiently considered: It is unclear if the topic of confounding by body habitus has been sufficiently considered. Is BMI a good measure of relative body fat composition? There are several articles showing that Serum PFAS are inversely correlated with body fat.**

Although not all studies show the same thing, the best performed studies in the most relevant age group suggest that some PFAS, notably the alkylates such as PFOA and PFNA are inversely associated with measures of body fat. (See for example Lind et al., PMID: 35074350 DOI: 10.1016/j.envres.2022.112677)

This is a possible confounder in terms of noncausal association (The data about body fat and PFAS are squishy). It can be addressed within the data for % body fat if available, a more specific measure than BMI or else mentioned as a possible study limitation that pertains to this work (and to many PFAS studies).

Reply: We do not have the possibility to measure percentage body fat, however we can estimate it using the formula presented by Deurenberg P, et al, 1991. This formula uses BMI, age, and sex to estimate the percentage body fat. However, since the models in our analysis already adjust for sex, age, and BMI, replacing BMI with an estimated percentage body fat will not affect the results (see example below). However, we agree that the lack of percentage body fat measurements should be included in the limitations of this study (page 30, lines 590-592).

“Another limitation is the lack of data regarding body fat, some PFAS have demonstrated an inverse association with body fat measurements specifically in women^{61,73}.”

Example: Logistic regression

Substance (ng/ml)	Using BMI				Using estimated % body fat			
	Base model		Model 1		Base model		Model 1	
	OR (95% CI)	P	OR (95% CI)	P	OR (95% CI)	P	OR (95% CI)	P
PFHxS	0.985 (0.923 - 1.051)	0.649	0.971 (0.902 - 1.047)	0.444	0.985 (0.923 - 1.051)	0.649	0.971 (0.902 - 1.047)	0.444
PFHpS	1.046 (0.979 - 1.119)	0.187	1.051 (0.975 - 1.132)	0.193	1.046 (0.979 - 1.119)	0.187	1.051 (0.975 - 1.132)	0.193
PFOA	1.039 (0.953 - 1.134)	0.381	1.050 (0.951 - 1.158)	0.334	1.039 (0.953 - 1.134)	0.381	1.050 (0.951 - 1.158)	0.334
PFOS	1.011 (0.945 - 1.083)	0.744	1.044 (0.967 - 1.126)	0.270	1.011 (0.945 - 1.083)	0.744	1.044 (0.967 - 1.126)	0.270
FOSA	1.035 (0.954 - 1.124)	0.410	1.066 (0.971 - 1.170)	0.181	1.035 (0.954 - 1.124)	0.410	1.066 (0.971 - 1.170)	0.181
N-MeFOSAA	1.026 (0.960 - 1.096)	0.458	1.038 (0.962 - 1.120)	0.341	1.026 (0.960 - 1.096)	0.458	1.038 (0.962 - 1.120)	0.341
N-EtFOSAA	1.040 (0.974 - 1.111)	0.238	1.031 (0.956 - 1.111)	0.428	1.040 (0.974 - 1.111)	0.238	1.031 (0.956 - 1.111)	0.428
PFNA	0.995 (0.921 - 1.074)	0.891	1.021 (0.937 - 1.113)	0.637	0.995 (0.921 - 1.074)	0.891	1.021 (0.937 - 1.113)	0.637
PFDA	0.982 (0.914 - 1.056)	0.628	0.986 (0.909 - 1.071)	0.742	0.982 (0.914 - 1.056)	0.628	0.986 (0.909 - 1.071)	0.742
PFDS	0.996 (0.941 - 1.058)	0.891	1.004 (0.939 - 1.074)	0.901	0.996 (0.937 - 1.058)	0.891	1.004 (0.939 - 1.074)	0.901
PFUnA	0.994 (0.933 - 1.060)	0.866	1.013 (0.942 - 1.090)	0.722	0.994 (0.933 - 1.060)	0.866	1.013 (0.942 - 1.090)	0.722
PFDoA	0.953 (0.876 - 1.036)	0.256	1.017 (0.923 - 1.120)	0.738	0.953 (0.876 - 1.036)	0.256	1.017 (0.923 - 1.120)	0.738
PFTTrDA	0.971 (0.930 - 1.014)	0.182	0.994 (0.946 - 1.045)	0.817	0.971 (0.931 - 1.014)	0.182	0.994 (0.946 - 1.045)	0.817
8:2 FTS	1.024 (0.979 - 1.070)	0.300	1.028 (0.977 - 1.082)	0.293	1.024 (0.979 - 1.070)	0.300	1.028 (0.977 - 1.082)	0.293
4-OH-CB187	1.078 (1.024 - 1.137)	0.005	1.081 (1.019 - 1.148)	0.010	1.078 (1.024 - 1.137)	0.005	1.081 (1.019 - 1.148)	0.010
3-OH-CB153	1.045 (1.005 - 1.086)	0.026	1.055 (1.011 - 1.101)	0.015	1.045 (1.005 - 1.086)	0.026	1.055 (1.011 - 1.101)	0.015
4-OH-CB107	0.988 (0.942 - 1.036)	0.616	0.989 (0.937 - 1.043)	0.674	0.988 (0.942 - 1.036)	0.616	0.989 (0.937 - 1.043)	0.674

- 2. Not considered: In muscle wasting, serum albumin can be lower. MS is a well-known cause of muscle inactivity and wasting. This is not just a theoretical consideration. In meta-analysis, serum albumin is inversely correlated with MS PMID 32982663 PMCID PMC7479227**

In contrast, multiple studies show that serum PFAS are directly correlated with serum albumin. (This is so well-known that references are not provided).

Together, these data present a potential cause of reverse causation. The authors may wish to consider if serum albumin could provide alternative, non-causal explanations for associations. My thought is that the associations could easily be robust to such considerations, yet there is no knowing if the questions are not asked and confidence in the study and its design will be higher if they are asked. (A simpler alternative is to only mention that this type of noncausal reason for association was not addressed, and could be a source of reverse causation, a problem also mostly missed by at least one of the cited references for another disease state(39)).

Reply: We have now measured serum albumin in all 1,814 individuals included in the study. We performed a Cox proportional hazard analysis analyzing albumin's effect on confirmed disability worsening. No association was observed between albumin concentrations and confirmed disability worsening (page 22, lines 364-369). The Cox proportional hazard analysis assessing PFAS and OH-PCBs association with confirmed disability worsening has now also been adjusted for serum albumin concentrations (page 22, lines 369-374). The inverse associations remained also after adjusting for serum albumin concentrations, indicating that these effects were not due to reverse causation. Moreover, 3-OH-CB153 showed a significant association with confirmed disability worsening in males, an association that previously was borderline significant.

“Albumin has been shown to be the major PFAS carrier in human blood^{37,38}. Moreover, a meta-analysis demonstrated lower concentrations of serum albumin in people with MS compared to HC subjects³⁹. This could potentially present a case of reverse causation. Thus, we further investigated if serum albumin concentrations affected the risk of confirmed disability worsening using Cox proportional hazard analysis. The results

demonstrated no association between serum albumin concentrations and risk of confirmed disability worsening (Supplementary Tables S41)."

"Moreover, when incorporating serum albumin concentrations in the model evaluating PFAS and OH-PCBs association with confirmed disability worsening, the inverse associations remained. Further, 3-OH-CB153 showed a significant positive association with confirmed disability worsening in males only (HR = 1.130, 95% CI: 1.010 - 1.260, and P = 0.035) (Supplementary Tables S42 to S44), thus indicating that the inverse associations observed are not due to reverse causation."

- 3. Insufficiently considered: Matching on current residence at diagnosis makes sense for serum PFAS, but it is imperfect for MS if there are immigrant participants. Foreign immigrants will generally come from lower (nearer equator) latitudes and have lower a priori risk for MS than those born in Sweden. (This source of potential bias is not addressed by measures of vitamin D at the time of diagnosis.)**

In addition, there are multiple demonstrations of different serum PFAS in specific ethnic populations, for example as seen in NHANES data across race/ethnicities and as seen in serum PFAS for residents of different countries. If there are immigrants in the study (not discernable from data I saw), they could be a priori more likely to be controls, and immigrants could have different serum PFAS than native populations (as shown in data from other nations). This might lead to a different type of bias, again depending on the actual number of immigrants in the study population. A sensitivity test could exclude foreign-born cases and controls.

Reply: We agree that immigration status is an important factor that should be taken into consideration. To address this issue we have added information concerning whether or not the participants are born in Sweden. Individuals born in Sweden to parents who are not immigrants are classified as Swedish. Based on this definition, 760 controls and 720 individuals with MS are considered Swedish. Moreover, individuals born in a Nordic country (Sweden, Norway, or Denmark) to parents who have not immigrated from outside the Nordic region are classified as Nordic. According to this, 789 controls and 760 people with MS are identified as Nordic. Finally, 43 females and 20 males, all with MS, did not

specify their country of birth. Therefore, immigrants are not overrepresented in the control group.

A sensitivity analysis has been performed demonstrating no effect on the risk of developing MS, independent of being native-born or not (page 17, lines 306-307). In the linear mixed-effect regression analysis, comparing PFAS and OH-PCBs concentrations in people with MS and controls, the p-values were affected mainly in females however similar trends were still observed. Due to missing values and the paired comparisons a total of 86 females and 40 males were excluded from the analysis, potentially explaining why the p-values were mainly affected in females (page 15, lines 280-282).

“A sensitivity analysis was performed demonstrating similar results when adjusting for Nordic heritage instead of Swedish (Supplementary Tables S26 to S29).”

“Sensitivity analysis demonstrated similar trends when adjusting for Swedish or Nordic-born (Supplementary Tables S7 to S14) as well as when pairing was removed, thus excluding the matching on residential area (Supplementary Tables S15 to S18).”

Of these three potential sources of bias, the reviewer considers the albumin issue to be the most important. As MS worsens, and if the meta-analysis data are representative of the actual relationship of MS to serum albumin, reverse causation could be an alternative explanation for key findings.

Abstract: The abstract is well written.

- 1. The authors could help the reader understand the point of the abstract better by clarifying syntax. It can be confusing to think about the inverse of worsening associated with a higher exposure (a double negative linked to a positive). From the abstract alone, the reviewer thinks but was not initially sure that the language means that higher serum PFAS is associated with better MS results over time (less disability). I needed to run to the tables to be sure my reading of the abstract was correct. Could the authors find some simple language solution so that the reader is not faced with solving the syntax problem of inverse to something bad (better or at least less awful outcome) associated with more internal exposure?**

Reply: We agree that the abstract should be rephrased using more straightforward language (page 3, lines 40-45). We have now also shortened the abstract in accordance with author guidelines.

“Moreover, two OH-PCBs (4-OH-CB187 and 3-OH-CB153) were associated with a significant ($P < 0.05$) increased risk of developing MS, regardless of sex and immigration status. With a clinical follow-up time of up to 18 years, increased serum concentrations of PFOA, PFOS, and PFDA significantly ($P < 0.05$) decreased the risk of confirmed disability worsening in both sexes, as well as PFHpS and PFNA, only in males with MS.”

- 2. If PCBs are an important part of the study, and it is not so clear, then they should be part of the abstract.**

Reply: PCBs are an important part of this study and we agree that it has not been made clear. To address this issue, we have added PCBs and the results to the abstract (page 3, lines 40-42).

“Moreover, two OH-PCBs (4-OH-CB187 and 3-OH-CB153) were associated with a significant ($P < 0.05$) increased risk of developing MS, regardless of sex and immigration status.”

Introduction: The introduction is mostly well written.

- 1. Line 99 Quoting: Moreover, low vitamin D levels have also been associated with an increased risk of MS.^{7,8}**

The reviewer's impression is that this hypothesis remains controversial. It is fine to mention it, but some acknowledgement of uncertainty is suggested, unless the authors can make an argument for definitive data. Other risk factors currently have wider support, especially wider distribution of age of first encounters with common infectious diseases, which is also considered to be an alternative explanation (a more important explanation than vitamin D) for the childhood-latitude distribution of MS.

Reply: We agree that other risk factors currently have wider support and that there is some uncertainty regarding vitamin D levels as a risk factor for MS. To address this comment we have clarified the uncertainty of this hypothesis in the quoted sentence (page 6, lines 104).

“Moreover, low vitamin D levels have also been associated with an increased risk of MS, however some uncertainty remains^{7,8}”

2. **Line 118 Quoting: the current knowledge and findings of PFAS associations with autoimmune disease are scarce and, at large, often have contradicting results in humans¹⁴**

This is explicitly a paper about a potentially autoimmune condition, so the reader may expect something more thorough. There is quite a bit of information about ulcerative colitis, for example, and increasing information about pregnancy-associated hypertension/preeclampsia (a specific setting). For these, and most evidence suggests an association of the condition to PFOA exposure and the concern that the PIH is all about reverse causation seems to be dispelled). For autoimmune thyroiditis, the data are complex and hard to interpret.

Reply: We agree that there is relevant literature on autoimmune diseases that should be included. We have now added more information regarding the associations between PFAS and ulcerative colitis and rheumatoid arthritis (page 7, lines 124-135).

“PFOA exposure has been associated with an increased risk of developing ulcerative colitis²⁷. In addition, a recent study demonstrated an association between higher PFAS concentrations and a decreased risk of developing rheumatoid arthritis²⁸. To this date, no risk analysis has been performed concerning PFAS exposure and the risk of developing MS. A cross-sectional study examining the relationship between autoimmune disease and estimated past exposure levels of perfluorooctanoic acid (PFOA), a common PFAS, demonstrated no significant association between PFOA levels and the incidence of MS²⁷. However, the PFOA concentrations were estimated from intake of contaminated drinking water, assuming low background exposure, and not measured. Additionally, a case-control study showed lower concentrations of PFOA in individuals with MS compared with controls, suggesting that PFOA exposure was not an important risk factor for MS, but no risk analysis was performed²⁹. To date, there are no studies investigating whether concentrations of PFAS have any role in disease progression in MS.”

3. **Line 122 Quoting Polychlorinated biphenyls (PCBs), another type of endocrine-disrupting chemicals, are, by contrast, well established as carcinogens and are known to have other harmful effects on human health, including developmental toxicity, neurotoxicity, and immunotoxicity. Consequently, many countries have restricted or banned their production and use since the 1970s, though they persist in the environment^{15,16}. In humans, hydroxylation of PCBs can occur, resulting in OH-PCBs, and the toxicity of PCBs has been suggested to originate partly from their hydroxylated metabolites. Moreover, OH-PCBs have a range of toxic traits not observed in parent PCBs, such as inhibition of mitochondrial respiration, disruption of thyroid hormone activity and estrogenic activity, and generation of reactive oxygen species¹⁶.**

The purpose of the paragraph about PCBs only is not that clear and the reasonable interpretation of these sentences and the thought about “in contrast” is that PFAS are not known carcinogens. IARC considers PFOA a class 1 and PFOS a class 2a carcinogen.

Reply: The idea behind the sentence was not that PFAS are not carcinogens, since as the reviewer states PFAS are considered as carcinogens. We agree that the “by contrast” leads to misinterpretation and to address this issue, it has been removed (page 7, lines 137).

“Polychlorinated biphenyls (PCBs), another type of endocrine-disrupting chemicals, are well-established as carcinogens and are known to have other harmful effects on human health, including developmental toxicity, neurotoxicity, and immunosuppression.”

4. **Line 131: The primary aim of this matched case-control study was to investigate the concentrations of PFAS and OH-PCB compounds in individuals recently diagnosed with MS and their matched controls (HC). Additionally, we aimed to explore whether the concentrations of PFAS and OH-PCB compounds influenced the course of the disease. We investigated the association between these chemicals and vitamin D3 levels and explored if the presumed protective effects of vitamin D3 are linked to PFAS and OH-PCB exposure.**

The abstract and introduction should be aligned. The abstract does not hint that PCBs are a crucial part of the MS consideration. Are they?

Reply: PCBs are an important group of endocrine-disrupting chemicals with a wide range of well-established harmful effects on human health. As such, PCBs are an important part of the MS consideration, and we agree that this should be reflected in the abstract. We have therefore included PCBs and the related findings in the abstract (page 3, lines 40-42 and 48-50).

“Moreover, two OH-PCBs (4-OH-CB187 and 3-OH-CB153) were associated with a significant ($P < 0.05$) increased risk of developing MS, regardless of sex and immigration status.”

“These results show previously unknown associations between OH-PCBs and the risk of developing MS, as well as the inverse associations between PFAS exposure and the risk of disability worsening in MS.”

Methods

1. **The methods come after the discussion. Time-wasting for reviewers and readers. That may be a feature of this journal and not a bug of this paper, in which case the authors have merely complied with anachronistic requirements.**

Reply: Since this is a feature of this journal we cannot make any changes to the structure.

2. **The use of BMI at two points in time is a valuable approach. However, noted above, relative body fat could change rapidly in early disease and BMI need not reflect that well in a disease state characterized by muscle-wasting. It appears possible to likely that relative body fat is inversely related to serum PFAS. IF there is no way to address this issue (it is rarely addressable with available data), it can be mentioned as a study limitation.**

Reply: This issue can, unfortunately, not be addressed with the available data. The best we can do is to estimate the percentage body fat using the formula presented in Deurenberg P, et al, 1991, which is based on age, sex, and BMI. However, since these three factors already are included in the model an estimate of percentage body fat instead of BMI will not change the results. See example in main issue 1. The lack of data regarding body fat has been included as a study limitation (page 30, lines 590-592).

“Another limitation is the lack of data regarding body fat, some PFAS have demonstrated an inverse association with body fat measurements specifically in women^{61,73}.”

- 3. The potential impacts of immigration on serum PFAS are partially addressed with the mononucleosis question, but it is the greater likelihood of absence of such diagnosis that is relevant for immigrants, who may be overrepresented in controls to the degree they are in the study. This deserves a sensitivity test if possible.**

Reply: We agree that immigration status is an important factor that should be taken into consideration. To address this issue we have added information concerning if the participants are born in Sweden or not. Individuals born in Sweden to parents who are not immigrants are classified as Swedish. Based on this definition, 760 controls and 720 individuals with MS are considered Swedish. Moreover, individuals born in a Nordic country (Sweden, Norway, or Denmark) to parents who have not immigrated from outside the Nordic region are classified as Nordic. According to this, 789 controls and 760 people with MS are identified as Nordic. Finally, 43 females and 20 males (all with MS) did not specify their country of birth. Therefore, immigrants are not overrepresented in the control group.

A sensitivity analysis has been performed demonstrating no effect on the risk of developing MS (page 17, lines 306-307). In the mixed-linear regression analysis, comparing serum concentrations in people with MS and controls, the p-values were affected however similar trends were still observed. This was mainly observed in females. Due to missing values and the paired comparison a total of 86 females and 40 males were excluded from the analysis, potentially explaining why the p-values were mainly affected in females (page 15, lines 280-282).

“A sensitivity analysis was performed demonstrating similar results when adjusting for Nordic heritage instead of Swedish (Supplementary Tables S26 to S29).”

“Sensitivity analysis demonstrated similar trends when adjusting for Swedish or Nordic-born (Supplementary Tables S7 to S14) as well as when pairing was removed, thus excluding the matching on residential area (Supplementary Tables S15 to S18).”

- 4. MS and albumin (inversely associated with MS in meta-analysis and directly associated with serum PFAS for obvious reasons of protein binding) is a reverse causation topic mentioned above. Because the logical reason for this finding is disease state, this may also be pertinent to disease course and not just to initial diagnosis.**

Reply: We have now measured serum albumin concentrations in all 1,814 persons. Albumin concentrations demonstrated no association with confirmed disability worsening (page 22, lines 364-369). Moreover, adjusting for albumin in the Cox proportional hazard analysis did not impact the inverse association observed between PFAS compounds and confirmed disability worsening (page 22, lines 369-374). (See main issue 2)

“Albumin has been shown to be the major PFAS carrier in human blood^{37,38}. Moreover, a meta-analysis demonstrated lower concentrations of serum albumin in people with MS compared to HC subjects³⁹. This could potentially present a case of reverse causation. Thus, we further investigated if serum albumin concentrations affected the risk of confirmed disability worsening using Cox proportional hazard analysis. The results demonstrated no association between serum albumin concentrations and risk of confirmed disability worsening (Supplementary Tables S41).”

“Moreover, when incorporating serum albumin concentrations in the model evaluating PFAS and OH-PCBs association with confirmed disability worsening, the inverse associations remained. Further, 3-OH-CB153 showed a significant positive association with confirmed disability worsening in males only (HR = 1.130, 95% CI: 1.010 - 1.260, and P = 0.035) (Supplementary Tables S42 to S44), thus indicating that the inverse associations observed are not due to reverse causation.”

Results:

The results are reasonably presented, interesting, and plausible. Overlooked implications that could alter their presentation follow under discussion.

1. The importance and relevance of the short-chain short half-life serum PFAS values is not that clear. The range is narrow, and the comparisons appear forced to this reviewer. If they are retained as an important part of the results and discussion, the limitations of relatively narrow ranges (and of course the frequent nondetects) should be more clearly discussed. This topic appears in the methods, and then it feels like it is minimized in results and especially discussion.

Reply: We do agree with the reviewer that the short half-lives of the short-chain PFAS (PFBS, PFPeS, PFHpA) make the assumption that the current concentrations reflect the exposure prior to diagnosis incorrect. The results of the linear regression (results presented in Figure 4/Supplementary Table S4-5) and the Cox proportional hazards models (Figure 5, Supplementary tables S35-S37) are still interesting as no such assumptions have been made. For the logistic regression, we chose to omit the following PFAS; PFBS, PFPeS and PFHpA due to their short half-lives in humans and the difficulties in drawing any conclusions on previous exposure based on the sample taken after diagnosis.

Furthermore, to clarify how the non-detects were handled in the analyses, we have added descriptions of that in the Result section (page 8, lines 175-176).

“Values below the detection limit were imputed with half the lowest measured concentration for each compound.”

Discussion:

- 1. 352-356, 428-430 Precise age-year specific serum PFAS are relevant to the diagnosis of MS, as the authors note, and are available in an NHANES study from age 12 onwards, which might assist readers to understand the authors' point.**

In addition, since the sexes reconverge (at least for the four most common serum PFAS) at around age 52 it is possible that sex considerations do not matter for diagnoses at later ages. (Up to the authors if this is helpful).

Reply: We have tried to clarify the reasoning behind these observed sex differences in PFAS and OH-PCB concentrations, mainly between MS and controls (page 26, lines 453-458).

“Finally, we also found apparent sex-specific differences in PFAS and OH-PCB concentrations between individuals with MS and HC subjects. In our case-control study, males diagnosed with MS had lower concentrations of short-chain PFAS compounds compared to their matched controls, whereas females with MS displayed higher concentrations of long-chained PFAS compounds compared to their matched controls. These differences were more pronounced in both males and females when comparing those with progressive MS to controls.”

- 2. 435-437 457-59 Meticulous residential matching is indeed a strength for serum PFAS comparisons, but it does little for the possible effect of immigration on MS diagnosis and any differences in either serum PFAS or MS course in immigrants vs native-born. The authors are likely to be aware that Immigration is a very important topic for MS specifically, and it is not addressed in the paper. Mentioned above, a sensitivity test limiting comparisons to natives, or a demonstration that the contribution of immigrant participants is a priori de minimis, or a mention of the limitation is commended to the authors to improve the paper.**

Reply: We agree that immigration is a very important topic in the context of MS. The control group is not overrepresented by immigrants. Native-born participants with native-born parents were defined as Swedish. Resulting in 760 controls and 720 people with MS being Swedish. Moreover, if a person is born in a Nordic country (Sweden, Norway, or Denmark) with parents who have not immigrated from non-Nordic countries, they are defined as Nordic. According to this definition, 789 controls and 760 people with MS are Nordic. Moreover, 63 individuals (all people with MS) have not defined the country born in.

A sensitivity analysis was performed, adjusting for whether a person is native-born or not. The results demonstrated similar trends, however, the increased concentrations of long-chained PFAS in females with MS compared to HC subjects were not significant (Supplementary Tables S7 to S10).

- 3. The authors can consider whether the % body fat predicts both increased MS severity and also lower serum PFAS is as plausible as the reviewer suspects, and not fully adjusted by BMI.**

Reply: This issue, can unfortunately not be addressed with the available data. The best we can do is to estimate the percentage body fat using the formula presented in Deurenberg P, et al, 1991, which is based on age, sex, and BMI. However, since these three factors are included already are included in the model an estimate of percentage body fat instead of BMI will not change the results. See example in main issue 1. The lack of data regarding body fat has been included as a study limitation (page 30, lines 590-592).

“Another limitation is the lack of data regarding body fat, some PFAS have demonstrated an inverse association with body fat measurements specifically in women^{61,73}.”

4. 369-374 and also 415-16, 446-48 Please see previous discussion about albumin, and also consider hemoglobin (also a protein that binds to PFAS and predicts serum PFAS!!) The authors are generally aware of this topic as regards menstruation, but it is not accounted for in the study design and adjustments. To what degree are the findings reverse causation, including the disease severity findings? One of the cited articles (39) is from a group which consistently ignored this topic. Clinically, albuminuria is common in diabetics and hypertensives, and often one of the overlooked presenting findings in diabetes. Lower serum albumin can be present even in the absence of albuminuria. (The group responsible for reference 39 wrote about the protective effect of PFAS on diabetes, a conclusion now regarded as a cautionary tale of overlooked reverse causation. It was subsequently shown that this is due in whole or part to proteinuria and lower serum protein that follows from diabetes.) To the degree possible, it is worth considering if this same kind of reverse causation might affect MS which has a meta-analysis showing inverse association to serum albumin. (This is likely to be less strong than in diabetes, but it is the same topic, and it will be more convincing to readers aware of the topic if it is considered.)

Reply: We agree that this should be taken into consideration and have now adjusted for albumin concentrations in the Cox proportional hazard analysis. Albumin is not *per se* associated with the risk of confirmed disability worsening according to our results. Moreover, adjusting for albumin in the Cox proportional hazard analysis which assesses PFAS and OH-PCBs association with risk of confirmed disability worsening, did not alter the previously observed results (page 22, lines 364-374). See point 2 for a more detailed answer.

“Albumin has been shown to be the major PFAS carrier in human blood^{37,38}. Moreover, a meta-analysis demonstrated lower concentrations of serum albumin in people with MS compared to HC subjects³⁹. This could potentially present a case of reverse causation. Thus, we further investigated if serum albumin concentrations affected the risk of confirmed disability worsening using Cox proportional hazard analysis. The results demonstrated no association between serum albumin concentrations and risk of confirmed disability worsening (Supplementary Tables S41). Moreover, when incorporating serum

albumin concentrations in the model evaluating PFAS and OH-PCBs association with confirmed disability worsening, the inverse associations remained. Further, 3-OH-CB153 showed a significant positive association with confirmed disability worsening in males only (HR = 1.130, 95% CI: 1.010 - 1.260, and P = 0.035) (Supplementary Tables S42 to S44), thus indicating that the inverse associations observed are not due to reverse causation.”

Furthermore, we have rewritten part of the discussion regarding reference (39) and the possibilities for reverse causation in T1D (page 21, lines 480-483).

“Diabetes and albuminuria, an early sign of kidney damage in diabetes, have been associated with lower perfluoroalkyl acids levels⁴⁸. It has been suggested that albuminuria drives an enhanced excretion of perfluoroalkyl acids due to kidney function decline^{37,48}, thus posing a risk of reverse causation in terms of PFAS association with risk of developing T1D.”

- 5. 376-378 The association of PFAS to CVD is far more complex than presented and has several recent well-done longitudinal studies which provide a more nuanced view of this topic than the authors’ have presented. This topic can either be omitted or expanded to reflect the emerging longitudinal findings more accurately.**

Reply: We agree and have now excluded CVD from the discussion, and as in the introduction we have expanded the discussion about autoimmune diseases instead. (page 27, lines 486-487 and 491-4965).

“Furthermore, PFOA exposure has also been associated with an increased risk of developing ulcerative colitis²⁷.”

“Exposure to PCBs has previously been associated with RA only in females⁴⁹. Moreover, a long-term follow-up study demonstrated that exposure to PCBs during a long time-period significantly increased mortality in females with systemic lupus erythematosus (SLE). However, the mechanism by which PCBs affect the development of SLE has not been studied^{50,51}. In T1D, an inverse association between PCB 153 exposure and T1D incidences has been reported⁵².”

6. **391-96 The authors present vitamin D associations as definitively causal (they present contravening clinical trial data, and the arguments against causation include that and more). The language should be altered to better reflect the controversy, or else the authors can better defend their causal language while accounting for more of the opposing perspective.**

Reply: We agree that the language should be altered to better reflect the controversy surrounding vitamin D, which has now been done (page 29, lines 547-551).

“When adjusting for 25(OH)D3 levels, the previously observed inverse associations were in fact strengthened, and also now PFBS, PFNA and PFDoA demonstrated inverse association as well, thus suggesting that the inverse effects by PFAS on the disease course cannot be fully explained by presumed protective 25(OH)D3 levels.”

7. **415-416 The reviewer perspective is that this would be a reasonable place to discuss how and why the disease severity findings could relate, in whole or in part, to reverse causation based on protein binding to albumin and hemoglobin. In contrast, the same considerations might strengthen the possibility of initial causation, about which the authors are appropriately understated (and could mention more).**

Reply: We have now also adjusted the Cox proportional hazard analysis for serum albumin concentrations (page 22, lines 369-374). This did not impact the inverse associations observed between PFAS and confirmed disability worsening. A discussion about the results regarding albumin has been added (page 28, lines 523-530).

“Moreover, when incorporating serum albumin concentrations in the model evaluating PFAS and OH-PCBs association with confirmed disability worsening, the inverse associations remained. Further, 3-OH-CB153 showed a significant positive association with confirmed disability worsening in males only (HR = 1.130, 95% CI: 1.010 - 1.260, and P = 0.035) (Supplementary Tables S42 to S44), thus indicating that the inverse associations observed are not due to reverse causation.”

“A meta-analysis study has previously demonstrated decreased serum albumin concentrations in people with MS compared to HC subjects³⁹. Moreover, a recent study

demonstrated that higher serum albumin concentrations are associated with a reduced risk of MS⁵⁹. Due to PFAS protein-binding properties, albumin has been shown to be a major PFAS carrier in blood^{37,38,59}. Thus, we further investigated if this could potentially present a cause of reverse causation in terms of PFAS inverse association with confirmed disability worsening. Adjusting for serum albumin concentrations did not impact the inverse associations observed between PFAS compounds and confirmed disability worsening, thus indicating that observed associations were not due to reverse causation.”

Reviewer #3:

I read this paper with interest. The authors should be commended on the thorough attempt to tackle this complex question using a unique cohort. As they state, the contribution of PFAS to autoimmune disease (and immunity in general) is generally poorly understood, but has the potential to deepen our understanding around both these compounds and also disease pathogenesis.

I did however have some areas where I felt that further clarification was required.

- 1. That the concentrations of these substances are typically not independent of each other is an important point, and needs to be made more clearly in the results. I note that the authors state that this violates the assumptions for multiple testing, yet it is not clear from the methods how the inherent concerns around multiple testing of associated exposures has been dealt with. I thought that this needed more clarity, as with a large range of compounds tested and few significant results (with significance in both directions of effect), this becomes a crucial consideration. Following on from this, the finding that the direction of effect in results reaching statistical significance is not consistent between males and females, and between compounds becomes of some concern, and I did not feel this was developed in depth in the discussion. There also appear to be differences when considering susceptibility (MS vs HC) and severity (CDS), however I found the discussion on why this might be slightly limited.**

Reply: We agree that the fact that these compounds are typically not independent from each other should be mentioned in the results as well. We have, therefore, added information about this in the results section (page 8, lines 177-179).

“It is important to note that exposure to these compounds is not independent of each other, and humans are generally exposed to multiple PFAS or OH-PCBs rather than single compounds. “

Furthermore, regarding the results reaching significance differently between males and females in the Cox proportional hazard analysis, that is discussed on page 28, lines 517-521. PCB153 has been associated with decreased serum testosterone levels. Low

testosterone levels have been observed in males with MS and have been associated with worse disability measurements. This could potentially explain the positive association seen between 3-OH-CB153 and risk for confirmed disability worsening in males. However, we cannot exclude the risk for false positives.

“Increased serum PCB153 levels have been associated with decreased serum testosterone levels in males⁵⁶. Moreover, lower testosterone levels have been associated with disability in males with MS⁵⁷. In animal models, testosterone has been observed to have beneficial effects on both inflammatory and neuroprotective mechanisms (reviewed 2018)⁵⁸. This could potentially explain the positive association between 3-OH-CB153 and the risk for confirmed disability worsening observed in males.”

- 2. In the methods, I note that DMT was not included in a mediator of disability outcomes. I would be interested as to why this decision was taken when other potential mediators of confirmed disability worsening were included. Given that DMT likely has a major influence on outcomes, and that DMT access is likely subject to substantial variation, for me this would be an important exposure to consider.**

Reply: We agree that DMT likely has a major influence on outcome and is an important exposure to consider which is why the Cox proportional hazard analysis was adjusted for DMT use. We apologize if this was not made clear enough in the text.

- 3. Additionally, I note that in the methods it is stated that those patients without EDSS within 3m of sample onset were excluded from the Cox analysis looking at CDW. In total only 551 patients were included in the CDW analysis – were these participants demographically identical to the whole cohort? There is no data presented that I could see on the phenotypes of those included in this analysis.**

Reply: The demography of the participants included in the CDW analysis has now been included in Supplementary Table S34. The general and clinical characteristics of these participants are similar to the entire cohort.

- 4. Follow up time in the CDW cohort is also unclear. I note that it is stated that FU was up to 18 years. What was the mean, median range? How was differing FU time dealt with in the Cox model, as this will depend on the variation in FU times.**

Reply: The follow-up time for the entire cohort is up to 18 years, however for the CDW cohort the mean follow-up time for females with RRMS was 10.7 ± 3.5 and the median was 10.9 ± 3.5 years. For males with RRMS the mean and median follow-up times were 10.2 ± 3.7 and 10.5 ± 3.7 years respectively. The mean and median follow-up times for males with PMS were 9.0 ± 4.1 and 8.7 ± 4.1 years respectively. For females with PMS the mean follow-up time was 9.0 ± 4.1 years and the median 9.7 ± 4.1 years. These numbers, including mean and median number of hospital visits, have been included in Supplementary Table S34.

Cox proportional hazard analysis is built to handle different follow-up times for different individuals. Cases where no events happen during the follow-up time are censored in this type of analysis. This should therefore not be an issue for the conducted analysis.

- 5. In the results and discussion, I would disagree with the assertion that these data show an effect of 25(OH)D3 on CDW. They show an association, which may result from reverse causation (those with worse disability have less sunlight exposure due to disability/behavioural change). The language around association vs causation is appropriate through most of the paper, but this did stand out.**

Reply: We agree and have now changed the language around association and causation in the context of 25(OH)D3 and CDW (page 25, lines 435-436).

“These results indicate that the PFAS inverse association with confirmed disability worsening cannot be fully explained by the levels of 25(OH)D3.”

Results

1. Table 1

I was unclear what % means for EDSS – is it the % for which one measure was available? Or the % with an EDSS available within a given time range of biosampling? I think that it was % with EDSS within 3m of biosampling, but this was only clear in the methods and not apparent when reading the table. One would normally give median rather than mean EDSS as this is a non-linear non-continuous measure.

Reply: We understand that there is a slight lack of clarity regarding what percentage means for EDSS. To address this, we have changed from percentages to the number of individuals where EDSS is available within the given time range (3 months from biosampling). This has also been clarified in a footnote to the table (page 10, line 184). Moreover, the median EDSS has also been added to the table.

“Description of EDSS measurements for individuals with an EDSS within three months from blood sample collection.”

2. **I note that EDSS is used rather than correcting for age with an ARMSS. Given the variation in both PFAS and EDSS with age is EDSS the right measure to use, or would applying an age correction be more rigorous?**

Reply: All analyses were made with correction for age. We preferred to do it this way rather than using a compound variable such as the ARMSS in order to have better control over the model.

3. **Only 13.7% of RRMS on treatment seems low – again is this at baseline? As there appears to be FU as part of this study this needs clarification and how graded exposure (in terms of both time and DMT efficacy) is dealt with.**

Reply: We are grateful to the reviewer for this comment. The given percentages in the table are at baseline, meaning that only 13.7% of the individuals with RRMS were undergoing some kind of treatment at baseline. Since this cohort includes relatively newly-diagnosed MS cases, treatment has not always been initiated at baseline.

In the Cox regression, we have taken into account the treatment at onset, but not taken into account changes in treatments and its duration during the follow-up time. An overly complex model increases the risk for overfitting, which is essential to avoid.

We agree with the reviewer that treatment during the follow-up time is an important factor in the context of risk for confirmed disability worsening. To address this issue, we have added three covariates in the model reflecting on whether the person has been treated with first-line DMT, second-line DTM, or other treatments during the follow-up time (page 19-20, lines 347-356).

“A sensitivity analysis was performed, adjusting the Cox proportional hazard analysis for treatment during the follow-up time. This strengthened PFOA and PFDA inverse association with confirmed disability worsening, while PFOS became borderline significant (HR = 0.906, 95% CI: 0.820 - 1.000, P = 0.053), independent of sex. Moreover, in males PFHpS, PFOS, and PFNA inverse association with confirmed disability worsening was strengthened, while PFOA became non-significant. Further, 3-OH-CB-153 demonstrated a positive association with confirmed disability worsening (HR = 1.130, 95% CI: 1.10 - 1.260, and P = 0.027). Finally, in females, the sensitivity analysis strengthened the inverse association between 4-OH-CB107 and confirmed disability worsening. Moreover, PFOA (HR = 0.820, 95% CI: 0.695 - 0.967, and P = 0.019) and PFDA (HR = 0.855, 95% CI: 0.750 - 0.975, and P = 0.019) also showed an inverse association with confirmed disability worsening (Supplementary Table S38 to S40).”

4. For irregular smoking, is this just of those who do not declare regular smoking?

Reply: Irregular smoking is for those who have stated that they smoked irregularly, and thus have not declared to be regular smokers.

5. The measure used to define alcohol consumption is not clear to me in the table, although I note it is given in methods.

Reply: We are grateful to the reviewer for this comment and agree that it might not be clear in the table. To address this issue, we have added a note in the table describing the measurement used to define alcohol consumption (page 10, lines 186).

“Alcohol consumption the week prior to sample collection has been recalculated into volume (cl) consumed 40% alcohol.”

6. Figure 3c

I found it difficult to appreciate the differences between regions given the scale on the graph. The low (highly significant) p values are were notable given the small differences illustrated. I note that multiple corrections were made but interestingly not disease status, although it appears that MS and HC were pooled? Could this have driven some of the findings?

Reply: We apologize for our mistake, the model was adjusted for the following covariates: age, sex, disease status, BMI at the age of 20, BMI at sample time point, year of sample collection, years of regular smoking, years of irregular smoking, years of indoor passive smoking, number of childbirths, and type of treatment. Moreover, the differences in the illustrations can appear small as they describe compound concentrations, while p-values have been adjusted for multiple covariates.

When looking into regional differences MS and HC have been pooled. However, this cannot have driven any findings since MS and HC are matched on residential areas, thus in each area the number of individuals with MS is equal to the number of HC.

7. Figure 5

Is 3-OH-Cb153 associated with positive association with disability worsening? It appears to be bordering on significance from the plot. This was not really mentioned in results that I could see.

Reply: It is true that 3-OH-CB153 has a positive association with confirmed disability worsening, bordering on significance. We have now included this in the results (page 19, lines 337-339) and also discussed these results in more detail (page 28, lines 509-510 and 517-521).

“Moreover, 3-OH-CB153 demonstrated a positive association with confirmed disability worsening in males, bordering on significance (HR = 1.110, 95% CI: 0.998 - 1.240, and P = 0.055).”

“Moreover, 3-OH-CB153 demonstrated a borderline-significant association with increased risk for confirmed disability worsening in males. “

“Increased serum PCB153 levels have been associated with decreased serum testosterone levels in males⁵⁶. Moreover, lower testosterone levels have been associated with disability in males with MS⁵⁷. In animal models, testosterone has been observed to have beneficial effects on both inflammatory and neuroprotective mechanisms (reviewed 2018)⁵⁸. This could potentially explain the positive association between 3-OH-CB153 and the risk for confirmed disability worsening observed in males.”

Point-by-point response to reviewers

We are sincerely grateful to all the reviewers for their comments, we believe that it has substantially improved this manuscript. The editorial requests have been addressed in the Author Checklist.

Reviewer #1

The revision is good. Could the title be clarified to be

From Disease Onset Risk to Disability Worsening not 'Disease Risk' which is too non-specific. Otherwise looks good.

Reply: To comply with the guidelines and clarify the title, we have changed the title from “*Associations between PFAS, OH-PCBs, and Multiple Sclerosis: From Disease Risk to Disability Worsening*” to “*Associations of PFAS and OH-PCBs with Risk of Multiple Sclerosis Onset and Disability Worsening*”.